# Bio-inspired vertebral design for scalable and flexible perovskite solar cells

Xiangchuan Meng [1,2], Zheren Cai [3], Yanyan Zhang [4], Xiaotian Hu [1,2 ✉], Zhi Xing [1], Zengqi Huang [1], Zhandong Huang[3], Yongjie Cui [5], Ting Hu [1,2], Meng Su [3], Xunfan Liao [5,6], Lin Zhang [7], Fuyi Wang [4], Yanlin Song[3 ✉] & Yiwang Chen [1,2,6 ✉]

The translation of unparalleled efficiency from the lab-scale devices to practical-scale flexible modules affords a huge performance loss for flexible perovskite solar cells (PSCs). The degradation is attributed to the brittleness and discrepancy of perovskite crystal growth upon different substrates. Inspired by robust crystallization and flexible structure of vertebrae, herein, we employ a conductive and glued polymer between indium tin oxide and perovskite layers, which simultaneously facilitates oriented crystallization of perovskite and sticks the devices. With the results of experimental characterizations and theoretical simulations, this bionic interface layer accurately controls the crystallization and acts as an adhesive. The flexible PSCs achieve the power conversion efficiencies of 19.87% and 17.55% at effective areas of 1.01 cm$^2$ and 31.20 cm$^2$ respectively, retaining over 85% of original efficiency after 7000 narrow bending cycles with negligible angular dependence. Finally, the modules are assembled into a wearable solar-power source, enabling the upscaling of flexible electronics.

[1] College of Chemistry, Nanchang University, 999 Xuefu Avenue, Nanchang 330031, China. [2] Institute of Polymers and Energy Chemistry, Nanchang University, 999 Xuefu Avenue, Nanchang 330031, China. [3] Key Laboratory of Green Printing, Institute of Chemistry, Chinese Academy of Sciences (ICCAS), 100190 Beijing, China. [4] CAS Key Laboratory of Analytical Chemistry for Living Biosystems, Institute of Chemistry, Chinese Academy of Sciences (ICCAS), Beijing 100190, China. [5] College of Materials Science and Engineering, Donghua University, Shanghai 201620, China. [6] Institute of Advanced Scientific Research (iASR), Jiangxi Normal University, 99 Ziyang Avenue, Nanchang 330022, China. [7] Hunan Key Laboratory of Super Microstructure and Ultrafast Process, School of Physics and Electronics, Central South University, Changsha 410083, China. ✉email: xiaotian@iccas.ac.cn; ylsong@iccas.ac.cn; ywchen@ncu.edu.cn

The flexible perovskite solar cells (PSCs) have triggered booming developments due to their superb photoelectric property, light-weight, low-cost, and feasibility in moderate-temperature roll-to-roll production[1–5]. So far, compared with the rigid devices as replacers for silicon-based solar cells, the flexible PSCs demonstrate extensively commercial potentials in the near future, such as wearable electronics, intelligent vehicles, and building-integrated photovoltaics. However, flexible PSCs face the translation from the lab-scale devices via spin-coating to the large-area modules by printing method with nonnegligible performance loss, which mainly hampers the practical applications[6–16]. In particular, the issues of perovskite crystal growth and defects passivation on flexible substrate are magnified when increasing the device area. Moreover, the fragility of indium tin oxide (ITO) as well as the perovskite crystals will be graver based on the large-area modules[17,18]. Therefore, scalable and robustly flexible perovskite solar cells require simultaneously enhance the quality of perovskite crystal and flexibility of the whole device.

As the anti-solvent treatment is not compatible with large-sized printing methods and the transport of perovskite solute is uneven for large-area perovskite (PVK) films, thereby the flatness, crystal nucleation, and growth are difficult to well control, especially upon the flexible substrate[19–21]. Recently, one approach to manipulate the rheological properties of the precursor ink is applied into large-area printing perovskite films. The additives of polymers, surfactants, and mixed solvents are validated can operate the printing dynamics and increase the adhesion to flexible substrate[22–25]. Another method to fabricate high-quality large-area perovskite films is to introduce some printing assisted processes, such as preheating substrate, air auxiliary and cover deposition, etc[26–29]. Meanwhile, the efforts about device design have also been reported to improve the flexibility of PSCs[17,18,30,31]. The alternatives of flexible transparent electrode for ITO, interfacial engineering, and additives for perovskite are three main directions. Although significant advances about large-area flexible PSCs have been made in crystal growth kinetics and flexible mechanics, most strategies only focus on one type of issue[31–33]. The "cask effect" of flexible PSCs is still a challenge for practicable applications. ITO electrode is usually the best conductive material to ensure device performance for the scalable PSCs. Unfortunately, due to the feature of brittleness, ITO electrode does not demonstrate perfect performance in flexible photoelectric devices. Although some flexible transparent electrodes such as silver nanowires and carbon nanotubes have shown the potential for application in flexible PSCs, they have not made outstanding progress in device efficiency, especially in the large-area devices[34–39]. Therefore, flexible PSCs based on the modified ITO electrode are also worth for further investigation.

In this work, we demonstrate a glued polymer interface layer (PEDOT:EVA, Poly(3,4-ethylenedioxythiophene):poly(ethylene-co-vinyl acetate)) due to the characteristics of both interaction with Pb atom, adhesiveness, and feasibility of synthetic PEDOT:EVA ink. In nature, vertebrae can adapt to complex human movements, because of the oriented crystallization of robust skeleton and the flexible structure with combination of hardware and software. Inspired by the biological crystallization and flexible structure of vertebrae[40–42], we synthesize the PEDOT:EVA ink by miniemulsion method, which has considerable dispersion and stability as the commercial PEDOT:PSS ink. More importantly, the PEDOT:EVA film yields perfect cohesion due to the adhesive EVA counterpart and acts as the hole transport layer (HTL) between ITO and perovskite films, which simultaneously facilitates perpendicular crystallization of perovskite on flexible substrate as well as sticks the brittle ITO and perovskite compactly to improve the flexibility. The resultant large-area (1.01 cm²) flexible PSCs are fully prepared by meniscus-coating, which achieve a stabilized efficiency of 19.87% with a strong mechanical stability.

In addition, due to the hydrophobic and packaged properties of EVA, the ion diffusion between perovskite and ITO films is also inhibited, retaining 85% of the initial efficiency after 3000 h under 1-sun illumination at room temperature. We further practically assemble the flexible PSCs into a module ($2 \times 2 \times 36 \, cm^2$) for wearable solar-power source, which is used to power the diversified electronics in a variety of body movements.

## Results

**Principles of bionic crystallization and structural design.** Social activities in a person's life are inseparable from the normal function of articular cartilage. Articular cartilage can protect the vertebrae from stress (σ) damage, one of the reasons is that the cartilage has the function of force absorption, which will distribute the force evenly and enlarge the bearing surface due to the elasticity and adhesiveness. This feature is similar to the PEDOT:EVA attribute, the mechanism of "vertebrae" bionics mainly comes from two aspects, bionic oriented crystallization and bionic structure (Fig. 1a). From the structural bionics, the PEDOT:EVA layer is applied between the brittle perovskite and ITO films, which acts as the cartilage between vertebraes to improve the mechanical flexibility of PSCs (the adhesiveness experiment of PEDOT:EVA is shown in Fig. 1b). As for the perspective of bionic crystallization, PEDOT:EVA layer accurately controls the nucleation sites and crystal oriented growth for high-quality flexible perovskite films (Fig. 1c). Therefore, based on the optimally synthetic conditions (see Supplementary Note 1), the excellent device efficiency can be ensured via the meniscus-coating technology (Fig. 1d, Supplementary Note 2), meanwhile, the PEDOT:EVA layer effectively absorbs and releases the stress to optimize the bending resistance for the flexible devices. Attribute to the synergistic optimization, high-quality large-area perovskite films can be fabricated with high repeatability and the perovskite module assembled by four pieces of single photoelectric device (36 cm²) is shown in Fig. 1e, which demonstrates the feasibility into the low-power devices (see Fig. 1e, Supplementary Fig. 1 and Supplementary Movies 1 and 2).

**Crystalline quality characterization and simulation.** In order to verify the feasibility of bionic crystallization, the flexible perovskite films prepared on PEDOT:EVA HTLs are characterized by various measurement techniques and dynamic simulation. The top-view scanning electron microscopy (SEM) images (Fig. 2a) reflect that the PEDOT:EVA layer promotes the crystallization of perovskite films and significantly increases the overall grain size. The average grain scale for perovskite films upon PEDOT:EVA is about 1.30 μm, which is much larger than 0.35 μm for the reference films (PEDOT:PSS-based perovskite films). And the corresponding atomic force microscope (AFM) images are shown in Supplementary Fig. 2. Besides an increase of grain size, the perovskite films based on PEDOT:EVA layer present smooth surface roughness, endowing the basis for enhanced performance of large-area PSCs. This consequence likewise can also be determined by 2D X-ray diffraction (2D-XRD) and XRD measurements (Fig. 2b and Supplementary Fig. 3). The relative intensity of (110) and (220) lattice planes of perovskite based on PEDOT:EVA layer become stronger and sharper, which further confirms the optimization of the perovskite crystallinity. Moreover, two bright spot-like patterns at 14.4° and 28.7° indicate the single crystalline-like uniaxial orientation feature for the perovskite crystals along the (110), (220) directions perpendicular to the substrate. On the contrary, for the perovskite films prepared on PEDOT:PSS layer, the clear ring-like patterns at (110) and (220) planes can be observed. The ring-like patterns mean that the grains exist as independent orientational domains, which is the typical feature of polycrystalline materials. Proving the

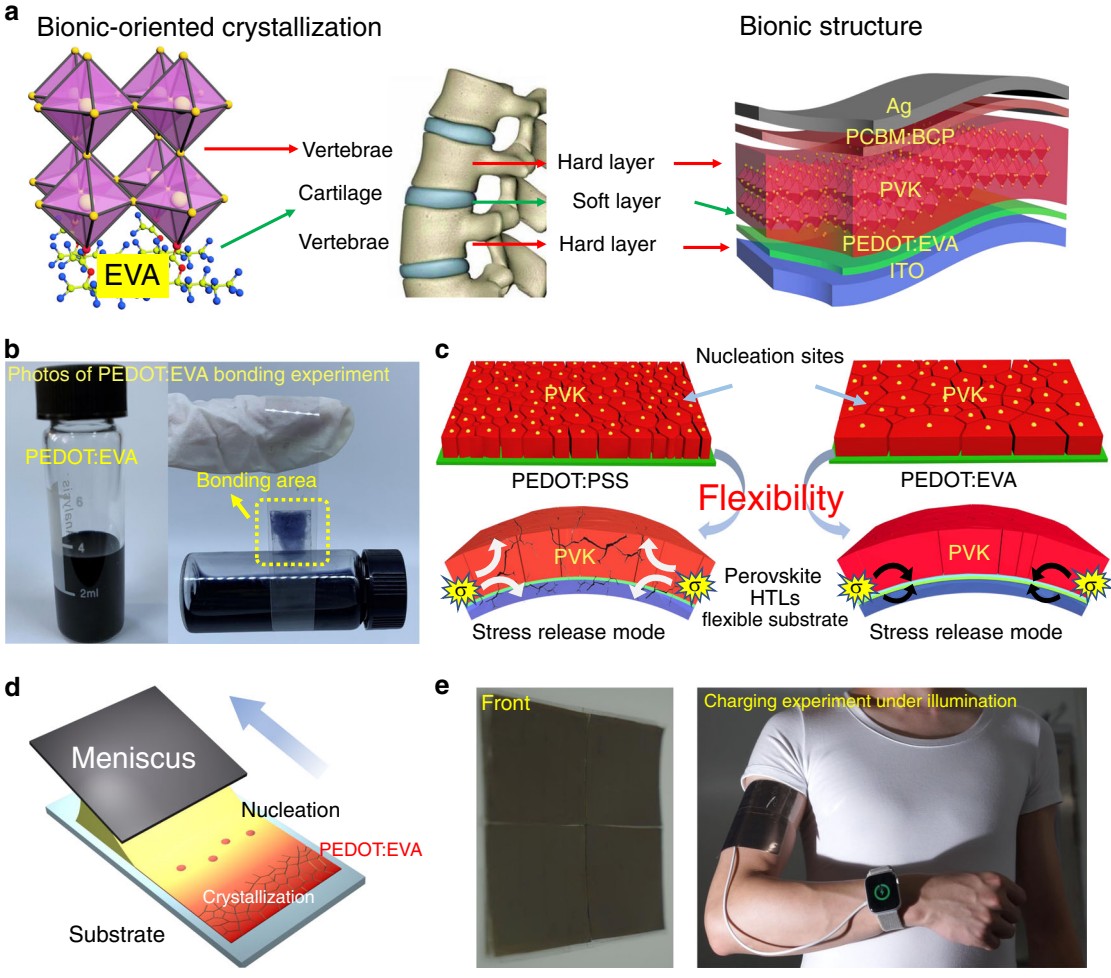

**Fig. 1 Schematic illustrations and photographs of the flexible PSCs. a** Biomimetic mechanisms of the vertebrae and PSCs. **b** Photographs of PEDOT:EVA bonding experiment. **c** Scheme of stress release for the PEDOT:EVA structure. **d** Schematic of PSCs by meniscus-coating. **e** Photographs of flexible perovskite solar modules as a wearable power source.

crystallization direction of perovskite crystal is random[43,44], this result corresponds to the observations from top-view and cross-sectional SEM images (Fig. 2a, Supplementary Fig. 4 and Supplementary Note 3), and also predicts superior spectral utilization.

To explore this bionic oriented crystallization, we speculate that grain growth process is strongly correlated with the nucleation sites at the interface (HTLs in this work). The effects on nucleation sites and nucleation rate can be well described by using the LaMer diagram (Fig. 2c)[45–48], which demonstrates three distinct regimes of nucleation and crystal growth for perovskite precursor solution on flexible substrates: (I) pre-nucleation, (II) nucleation and crystallization growth and (III) crystallization (see detailed explanations about crystallization kinetics in Supplementary Note 4). From the above three stages, the best way to optimize the quality of flexible perovskite films is to reduce the nucleation sites appropriately and extend the grain growth process, which has been verified by the Arrhenius type equations (Supplementary Eqs. 1 and 2). Based on these equations, the crystal free energy ($\Delta G_c$) is mainly adjusted by the surface energy of substrate. Therefore, we measured the corresponding surface energy value, and dynamic contact-angle measurements and schematic diagram are shown in Fig. 2c, Supplementary Fig. 5 and Supplementary Table 1, respectively. Compared with the reference, the PEDOT: EVA layer shows greater $\Delta G_c$, lower crystallization rate, and reduced nucleation site due to the higher surface energy. Meanwhile, shorter nucleation time also provides more time for grain

growth to ensure the quality of perovskite films, which corresponds to the results of SEM images and 2D-XRD measurements. In addition, UV–vis absorption spectrum, steady-state photoluminescence (PL) and time-resolved photoluminescence (TRPL) measurements are performed to directly confirm the quality of perovskite films (Fig. 2d–f). As expected, the perovskite films fabricated on PEDOT:EVA illustrate stronger absorption than the reference in the full UV–vis spectra, as well as showing smaller fluorescence quenching phenomenon from the forward and backward scanning and the shorter carrier recombination lifetime. The low and similar fluorescence intensity of the samples upon PEDOT:EVA layer indicates that the carrier defect concentration on the top and bottom surface can be effectively reduced, which will significantly optimize the pairing between surface defect and bulk defect caused by the thermodynamic instability states in the annealing process for perovskite films and decrease the non-radiative recombination loss. Furthermore, the perovskite films based on PEDOT:EVA layer yield shorter carrier recombination lifetime than reference (9.8 versus 14.6 ns, Supplementary Table 2). The PEDOT:EVA layer could effectively extract and transfer the charge carriers from perovskite films to anodes as compared with the PEDOT:PSS.

With this respect, the first-principles density functional theory (DFT) computational analysis and the Car-Parrinello Molecular Dynamics (CPMD) simulation of the interaction between MAPbI$_3$ and EVA using the slab model of both materials are

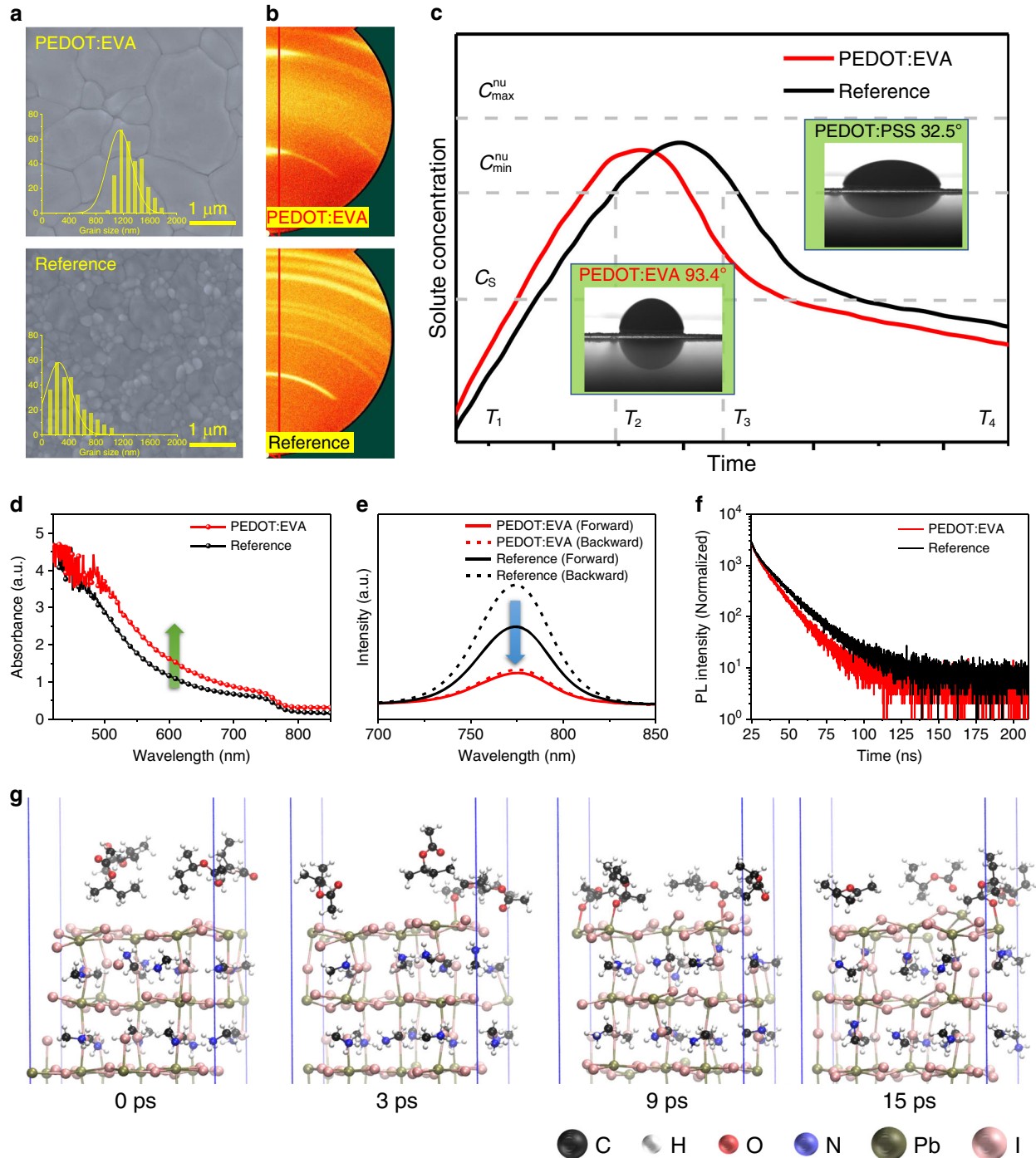

**Fig. 2 Crystalline quality characterization and crystallization kinetics simulation. a** SEM images, **b** 2D-XRD patterns, and **c** LaMer curves for perovskite film based on PEDOT:EVA or PEDOT:PSS (reference) substrates (the insets are the contact-angle measurements). **d** UV–vis absorption spectrum and **e** The steady-state photoluminescence spectra for the perovskite film. **f** Time-resolved photoluminescence decay for the perovskite film based on PEDOT: EVA and reference. **g** The Car-Parrinello Molecular Dynamics (CPMD) of PbI$_2$-terminated with EVA for the perovskite films.

further used to investigate the nature of perovskite crystallinity improvement (Fig. 2g, Supplementary Figs. 6–8, Movies S3–6 and Supplementary Note 5)[49–52]. The simulation model we choose is EVA molecular chain surface covered by (001) MAI−terminated MAPbI$_3$ units, and the system dynamics are evolved at 350 K for several picoseconds. Then, we observe the structural interface re-arrangement (0 ps). At the early stage of nucleation, Pb atoms will interact with the carboxyl group of EVA from PEDOT:EVA layer, thus replacing the interaction with sulfonic radical of PSS to produce crystal nuclei (during the first 15 ps). Afterward, MAI

and PbI$_2$ begin to grow into MAPbI$_3$ and undergo the crystallization growth process. In order to explain the specific reason of this early nucleation, we further monitored the binding energy of Pb atom with carboxyl group on EVA molecular and sulfonic radical on PSS molecular (Supplementary Fig. 9). Obviously, the binding energy of Pb atom to carboxyl group is much stronger than that of sulfonic radical, which means that the Pb atom in PbI$_2$ is more likely to bind with EVA molecular and the result is consistent with the DFT computational analysis and the CPMD simulation. The interaction between the carboxyl group of EVA

and Pb atom can also be confirmed in X-ray photoelectron spectroscopy (XPS) measurement. Due to the electronegativity of O atom is stronger than I atom, the interaction between O and Pb will lead to an increase in the binding energy of some Pb atoms' 4f orbitals[53]. Then, it makes the peak of Pb 4f shifting towards the high energy position, as shown in Supplementary Fig. 10, which implies the interaction between the carboxyl group of EVA with Pb atoms. Noteworthily, the carboxyl group on the surface of PEDOT:EVA is less than that of sulfonic radical on PEDOT:PSS substrate. Therefore, the former based on PEDOT:EVA layer has fewer nucleation sites, which is conducive to the increase of grain scale and crystallization quality. This conclusion is also consistent with the analysis of LaMer curves and free energy theory ($\Delta G_c$). Thus, based on the above discussion of crystallization optimization, we demonstrate that the PEDOT:EVA layer can interact with perovskite precursor ink to produce fewer perovskite crystal nuclei, and the integrated effect suffers sufficient crystal growth conditions for flexible perovskite films.

**Performance of flexible PSCs upon PEDOT:EVA interface.** Before the characterization for the PSCs based on different HTLs,

the intrinsic properties of PEDOT:EVA films are tested compared with the PEDOT:PSS HTLs and the results are shown in Supplementary Note 6, including the conductivity, optical transmittance, and film morphology. Then, to examine the advance of PEDOT:EVA HTLs, the ultraviolet photoelectron spectroscopy (UPS) and the corresponding energy level diagram of PSCs are shown in Fig. 3a and b, respectively. The work function of PEDOT:EVA is about −5.3 eV, which is much higher than that of PEDOT:PSS and closer to the valence band of perovskite. It is because of the incorporation of insulating EVA content in PEDOT[18]. The matched energy level alignment predicts an optimization of open-circuit voltage ($V_{oc}$) for PSCs, resulting that photoexcited holes from perovskite films will be effectively extracted and transferred to the anode. As shown in Fig. 3c, Mott–Schottky plots for PSCs upon PEDOT:EVA and reference are measured with the frequency of 10 kHz at the temperature of 300 K to investigate the interface properties between the HTLs and perovskite films by conducting capacitance-voltage (C-V) measurement, and the device structure is polyethylene glycol terephthalate (PET)/ITO/HTLs/perovskite/Au. In general, the $V_{oc}$ of devices is directly correlated to the flat-band potential

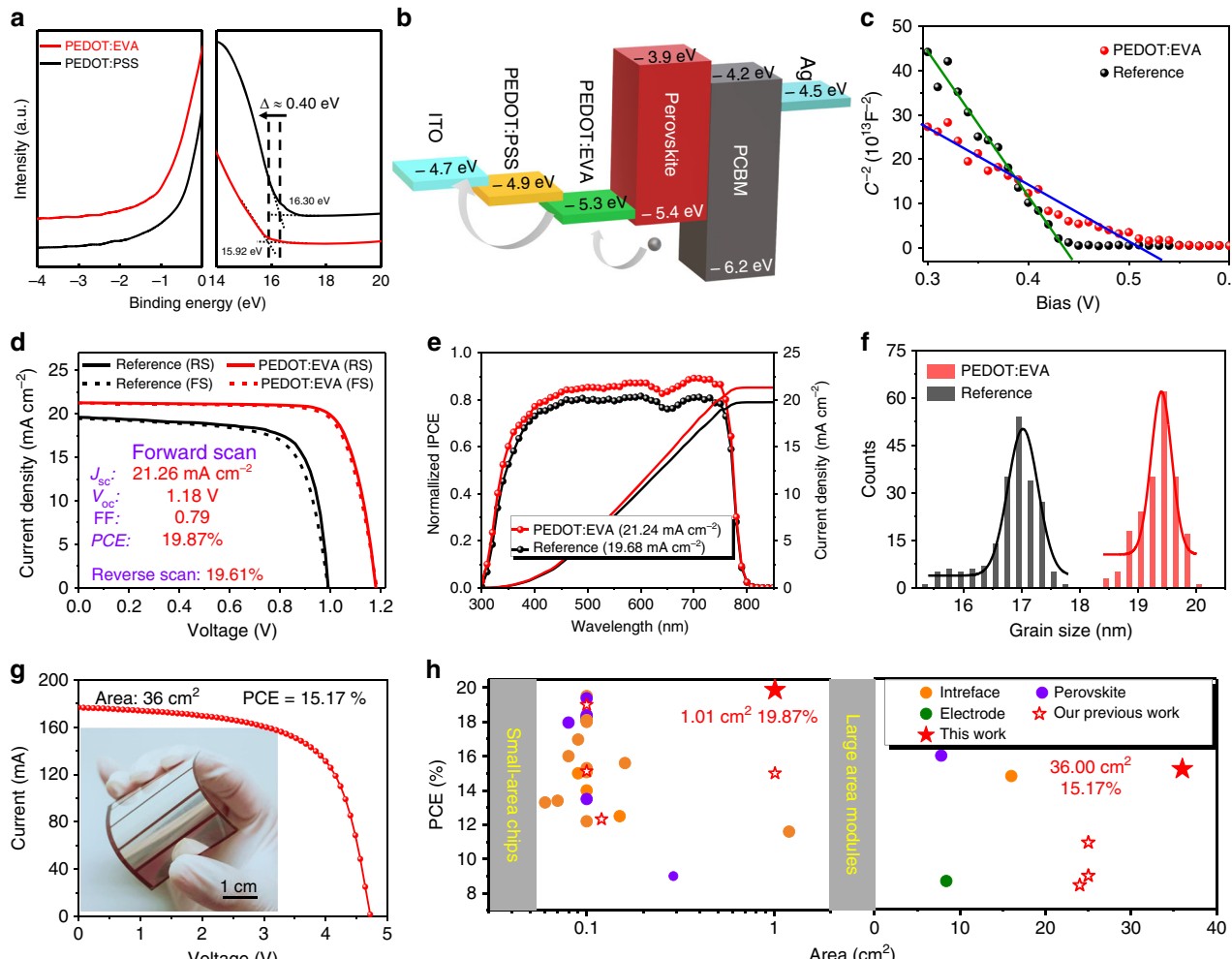

**Fig. 3 Performance of PSCs based on PEDOT:EVA and reference. a** Ultraviolet photoelectron spectroscopy (UPS) spectra for the PEDOT:EVA and PEDOT:PSS. **b** Energy level diagram of the perovskite solar cells (data for the PEDOT:PSS, PEDOT:EVA, and perovskite layers are provided by UPS measurement). **c** Mott–Schottky plots of PSCs. **d** J-V curves of the flexible PSCs measured in both the reverse and forward directions (the device effective area is 1.01 cm²). **e** The corresponding IPCE spectra of flexible PSCs. **f** Performance distribution of flexible PSCs. **g** I-V curves of the flexible PSMs based on PEDOT:EVA. **h** The summary of PCE and different effective of recently reported flexible PSCs. The orange dots, purple dots, and green dots represent highlights in interface, perovskite, and electrode, respectively. The red hollow stars represent our previous work and the red solid stars represent this work (In consideration of sufficient mechanical flexibility, we exclude the devices based on flexible metal and glass.).

**Table 1 Photovoltaic performance of the PSCs based on different flexible substrates.**

| Device | | $J_{sc}$ (mA cm$^{-2}$) | $V_{oc}$ (V) | FF | PCE (%) |
|---|---|---|---|---|---|
| Reference[a] | Reverse | 19.61 | 1.00 | 0.73 | 14.30 |
| | Forward | 19.46 | 1.00 | 0.70 | 13.63 |
| | Average | 19.40 ± 0.26 | 1.00 ± 0.01 | 0.68 ± 0.06 | 13.21 ± 1.12 |
| PEDOT:EVA[a] | Reverse | 21.26 | 1.18 | 0.79 | 19.87 |
| | Forward | 21.24 | 1.18 | 0.78 | 19.61 |
| | Average | 21.21 ± 0.05 | 1.18 ± 0.01 | 0.78 ± 0.02 | 19.52 ± 0.21 |
| PSMs[b] | Actual area | 4.92 ± 0.14 | 4.72 ± 0.01 | 0.65 ± 0.15 | 15.01 ± 0.18 |
| | Effectual area | 5.34 ± 0.18 | 4.73 ± 0.01 | 0.68 ± 0.21 | 17.20 ± 0.35 |

The average and standard deviation values are based on 50 cells and the "±" is defined as the error bar.
[a]The effective area for devices is 1.01 cm$^2$.
[b]The effective area for devices is 36.00 cm$^2$.

($V_{bi}$, Supplementary Note 7). The $V_{bi}$ value for the devices based on PEDOT:EVA is 0.53 V, which is higher than that based on PEDOT:PSS (0.44 V). The enhanced $V_{bi}$ value of 90 mV can contribute a sharp increase of $V_{oc}$ in the devices with PEDOT:EVA layer. Moreover, the doping density is also calculated by the Mott–Schottky equation (Supplementary Eq. 3, Supplementary Eq. 4 and Supplementary Note 7). Compared with the devices coated on PEDOT:PSS, a significantly increased doping density ($N_d$) and trap passivation effect in PEDOT:EVA based devices could improve the charge collection efficiency. The depletion zone reflects a bulk resistance from perovskite to HTL, which can be expressed as depletion zone width ($W_p$). For the devices based on PEDOT:EVA HTLs, the wider $W_p$ (12.141 versus 10.053 nm) means a better charge extraction capacity from perovskite to HTL[54,55].

By applying this bionic design, we test the photoelectric conversion efficiency (PCE) of flexible solar cells. A certified aperture is applied to define the effective area for the $J$ (current density)-$V$ (voltage) and $I$ (current)-$V$ curve measurements. Based on the flexible structure of PET/ITO/PEDOT:EVA/perovskite/PCBM/BCP/Ag, there is a substantial improvement in performance as compared with the reference. Detailed device parameters have been shown in Fig. 3d, Supplementary Fig. 13 and Table 1 (the average and standard deviation values of flexible PSCs are calculated by at least 50 cells and the "±" is defined as the error bar.). We achieve a $V_{oc}$ of 1.18 V, a fill factor (FF) of 0.79, and a stabilized PCE of 19.87% without evident hysteresis—among the highest efficiency reported in flexible PSCs (1.01 cm$^2$). The steady-state PCE and $J_{sc}$ of flexible PSCs are observed over 600 hours in Supplementary Fig. 11, which enables an excellent illuminated stability. We also apply this hole transport layer to rigid devices as for validating the feasibility of this bionic crystallization. The PCE of rigid PSCs achieves a PCE of 22.16% with a short-circuit current density ($J_{sc}$) of 22.91 mA cm$^{-2}$, a $V_{oc}$ of 1.18 V and a FF of 0.82 under reverse scan (Supplementary Table 3 and Supplementary Fig. 12), which shows better performance than that of rigid reference (17.61%, the effect of perovskite film thickness with different meniscus moving speed, meniscus spacing and VASP pressure on device performance is shown in Supplementary Fig. 13 and Supplementary Table 4). It is worth noting that all the $J_{sc}$ values are consistent with those integrated by the external quantum efficiency (EQE) spectra (Fig. 3e and Supplementary Fig. 14). Furthermore, we explore the superiority of PEDOT:EVA in charge carrier transport for PSCs by $J$-$V$ measurement under different illumination (Supplementary Fig. 15), dark $J$-$V$ curves (Supplementary Fig. 16), electrical impedance spectroscopy (EIS, Supplementary Fig. 17) and space-charge-limited-current (SCLC, Supplementary Fig. 18, Supplementary Table 5 and Supplementary Note 8), respectively. All of these are attested to the improvement

of PCE. In addition, the reproducibility of the device performance is further verified, fifty flexible PSCs upon PEDOT:EVA from five batches are prepared and the specific device parameters are summarized and listed in Fig. 3f, Supplementary Fig. 19 and Supplementary Table 6, which shows a superior reproducibility with the narrow PCE distribution from 18.73 to 19.87% as compared with the reference.

Encouraged by the above satisfactory flexible device efficiency and reproducibility, we fabricate the $6 \times 6$ cm$^2$ perovskite solar modules (PSMs) from four single sub-cells, the construction and photographs are shown in "Methods", Supplementary Fig. 1, Figs. 1e and 3g, respectively. The area of PSMs is 36 cm$^2$ (the effective area is 31.2 cm$^2$) with a geometric fill factor of 86.7%. The champion PCE is 15.21% based on the module area (17.55% based on the effective area), and the steady-state PCE, $V_{oc}$, $J_{sc}$ of encapsulated flexible PSMs and the certification efficiency (14.91%) are shown in Supplementary Figs. 20 and 21. The PSMs can output 308 mW maximum power over 1200 h. Similarly, fifty flexible PSMs upon PEDOT:EVA from five batches are also prepared and the specific device parameters are summarized and listed in Supplementary Table 7, and randomly selected 12 points from the PSMs are used to investigate the reproducibility. The corresponding UV–vis absorption and steady-state photoluminescence spectra of perovskite films deposited on PEDOT:EVA are measured and shown in Supplementary Fig. 22. In one piece of PSMs, the UV-via absorption spectra and fluorescence intensity of 12 test points selected randomly do not change obviously, which further verifies the performance and stability of the PSMs. The module result once again confirms that the large-scale high-quality flexible perovskite films based on PEDOT:EVA are prepared by meniscus-coating. The record performances of flexible PSCs and PSMs are achieved. To determine the efficiency, we summarize the recent representative reports of flexible PSCs with different approaches and effective areas[3–5,9,12,18,30–33,56–61] in Fig. 3h. The major highlights in the interface, perovskite, and electrode of flexible PSCs are marked as the color of cyan, blue, and orange, respectively. And the hollow stars and solid star represent our previous work and this work. Evidently, we demonstrate the record performances for flexible chips and modules. The satisfying PCE of large-area modules is the crucial touchstone for future applications. Finally, four above-mentioned PSMs are practically connected into series to assemble a flexible wearable solar-power source (Fig. 1d, Supplementary Movies 1 and 2).

**Mechanical and environmental stabilities for flexible PSCs.** Besides the photoelectric performance, mechanical, and environmental stability are also the main challenges to be solved in flexible PSCs and PSMs[62,63]. Here, the PEDOT:EVA is designed as a blocking glue, which bonds ITO and perovskite together, and

the cohesiveness of PEDOT:EVA ink is demonstrated by the strain-stress measurement (Supplementary Fig. 23. Compared with the PET plastic bonded with PEDOT:PSS, the PET plastic bonded with PEDOT:EVA exhibits significantly superior bonding performance, which can maintain mechanical stability until fracture occurs at a strain ratio of 17–18% (for the PEDOT:PSS, the fracture-tensile ratio is no >10%), confirming that the PEDOT:EVA ink has an optimistic cohesiveness[64,65]. The cohesiveness of EVA originates from the adsorption interaction of EVA emulsion (Van Der Waals or hydrogen-bonding interaction) in the PEDOT:EVA ink prepared by the miniemulsion method, which has been reported in previous adhesive work[66,67]. Therefore, the PEDOT:EVA layer accordingly becomes the center of stress absorption and release like the spinal cartilage, improving the mechanical toughness of photoelectric devices. Then, the contribution of PEDOT:EVA and PEDOT:PSS to the bending resistance are explored through the AFM images characterization for the ITO and perovskite films. For the ITO transparent electrode coated with PEDOT:EVA, no obvious micron-scale crack appears after 4500 cycles, which is far better than that coated with PEDOT:PSS. The integrated optimization of mechanical stability is explained by finite-element simulation, the calculation of strain energy release rate, mechanical mismatch coefficient, and the corresponding results are also shown in Supplementary Note 9, Supplementary Fig. 24 and Supplementary Fig. 25.

Meanwhile, in order to attest the stickiness of PEDOT:EVA, the SEM images for the perovskite films upon PEDOT:EVA and PEDOT:PSS layers after bending 7000 cycles are shown in Fig. 4a and Supplementary Fig. 26. No obvious cracks are found on the films based on PEDOT:EVA layer with a bending radius of 3 mm, while evident cracks are determined on the perovskite films upon PEDOT:PSS layer. The XRD patterns under different bending cycles intuitively reflect the bending resistance of perovskite films (Supplementary Fig. 27). For the films prepared on PEDOT:EVA, characteristic peak intensity and position for the perovskite films ((110), (220), and (310)) do not change significantly after 7000 bending times. However, for the perovskite films upon PEDOT:PSS, the intensity of the perovskite characteristic peak is gradually weakened. The characteristic peak of PET (near 26°) is evidently determined, indicating the devices have suffered serious damage. Furthermore, the Young's Modulus of the film surface for PET/ITO/PEDOT:EVA HTL is 139 MPa measured by AFM mechanical model, which is less than the value of PET/ITO/PEDOT:PSS[68,69] (258 MPa, Supplementary Fig. 28 and Supplementary Table 8). Based on the corresponding mechanical parameters, the finite-element method is used to simulate the stress distribution for the PSCs under bending (Fig. 4b, c and Supplementary Note 10). It is easy to find that PEDOT:EVA interface can significantly reduce the whole stress distribution of ITO and PVK films, which is consistent with the morphological and mechanical measurements.

Then, we measure the efficiency variations for PSCs at 7000 consecutive bending cycles with different bending radii ($R = 10$, 5, and 3 mm, respectively). And the detailed results can be seen in Fig. 4d, e, Supplementary Fig. 29). After 7000 cycles with 10 mm, 5 mm or 3 mm bending radius, the PCE of PSCs based on PEDOT:EVA substrate can still maintain about 96, 95 and 85% of the initial PCE, respectively. After the modification of PEDOT:EVA, the bending radii limit of ITO can reach 3 mm. However, the PCE for the devices upon PEDOT:PSS layer decreases significantly, only 51%, 10% and 2% of the original PCE can be retained, which directly proves the stickiness of PEDOT:EVA under bending resistance. This optimization of mechanical stability provides extreme universality for wearable applications. Therefore, the PCE of flexible PSCs is investigated under different

bending conditions (Fig. 4f, bending angle = 10°, 20°, 30°, 40°, 50°, 60°, 70° and 80°, respectively). The results are shown in Fig. 4g and Supplementary Table 9 (the corresponding actual photographs of cyclic bending test is shown in Supplementary Fig. 30). Meanwhile, in order to exclude the effect of effective area variation caused by bending, we revise the PCE value for the PSCs with different bending angle. Even if the bending angle reaches 60°, the revised efficiency can still maintain more than 19%. Only when the bending angle continues to increase, the PCE of PSCs will gradually decrease, which proves that the devices prepared on PEDOT:EVA have good bending ability without substantial angular dependence. To determine the mechanical performance, we also summary the recent representative reports of flexible PSCs with different approaches, and the results are shown in Supplementary Fig. 31 and Supplementary Table 10.

The long-time environment stability investigations in air are shown in Supplementary Fig. 32, the PSCs based on PEDOT:EVA still retains 85% of the original PCE after 3000 h, in comparison, the encapsulated devices upon PEDOT:PSS HTLs are precipitously degradable after 2450 h, which is due to the corrosion of acid PEDOT:PSS and the ion diffusion in ITO and perovskite films. To intuitively explore the mechanism for the improvement of environmental stability, we apply the time-of-flight secondary ion mass spectroscopy (ToF-SIMS) to probe into the element distribution with aging time (Fig. 4h). And the I$^-$, CN$^-$, and InO$^-$ are detected to characterize the perovskite in-depth profiles. Obviously, there is no significant I$^-$, CN$^-$ or InO$^-$ ion diffusion in the devices based on PEDOT:EVA substrate after 21 days of exposure to air. However, the I$^-$ in PVK and the InO$^-$ in ITO from the device based on PEDOT:PSS diffuse into ITO and PVK layer, respectively, which are related to the acidity and moisture absorption of PEDOT:PSS, as well as the nature of perovskite and ITO themselves. These evidences show that hydrophobic PEDOT:EVA (the pH value is 6.49, which is more optimal than 1.20 for PEDOT:PSS) can avoid the erosion of perovskite by acidic PEDOT:PSS as well as the ion diffusion between PVK and ITO. Thus, the robustly flexible and environmentally stable PSCs demonstrate great application potentials (the corresponding optical, electrical properties, and characteristics of PEDOT:EVA are shown in Supplementary Figs. 33-38).

## Discussion

Inspired by the orientated crystallization and flexible structure of the vertebrae, we introduce a PEDOT:EVA interfacial layer between the ITO and perovskite layers for large-area flexible PSCs. The non-wetting and glued PEDOT:EVA layer can simultaneously afford perpendicular crystal growth and enhanced flexibility of the whole device. Accordingly, a record efficiency of 19.87% for 1.01 cm$^2$ flexible PSCs with excellent mechanical stability has been achieved. Moreover, due to the highly reproducible and practicable performance, a 36 cm$^2$ module is fabricated and applied into the wearable solar-power source. These findings open up a general approach for crystalline materials in flexible and wearable electronics.

## Methods

**Materials**. All the chemical materials and reagents are purchased and used directly without further optimization or purification, including bathocuproine (BCP, Sigma-Aldrich), poly(ethylene-*co*-vinyl acetate) (EVA, vinyl acetate 40 wt%), anhydrous N, N-dimethylformamide (DMF, Sigma-Aldrich), anhydrous dimethyl sulfoxide (DMSO, Sigma-Aldrich), anhydrous ethyl alcohol (99.5%, Sigma-Aldrich), lead (II) iodide (PbI$_2$, 99.999% purity, Afar Aesar), polyurethane (PU, Sigma-Aldrich), lead(II) bromide (PbBr$_2$, Sigma-Aldrich), formamidinium iodide (HC(NH$_2$)$_2$I, Xi'an p-OLED Corp), aqueous PEDOT:PSS solution (CLEVIOS$^{TM}$ Al 4083, Heraeus), methylamine iodide (MAI) (>98% purity, Dyesol) and [6,6]-phenyl-C$_{61}$-butyric acid methyl ester (PC61BM, American Dye Source Inc), anhydrous

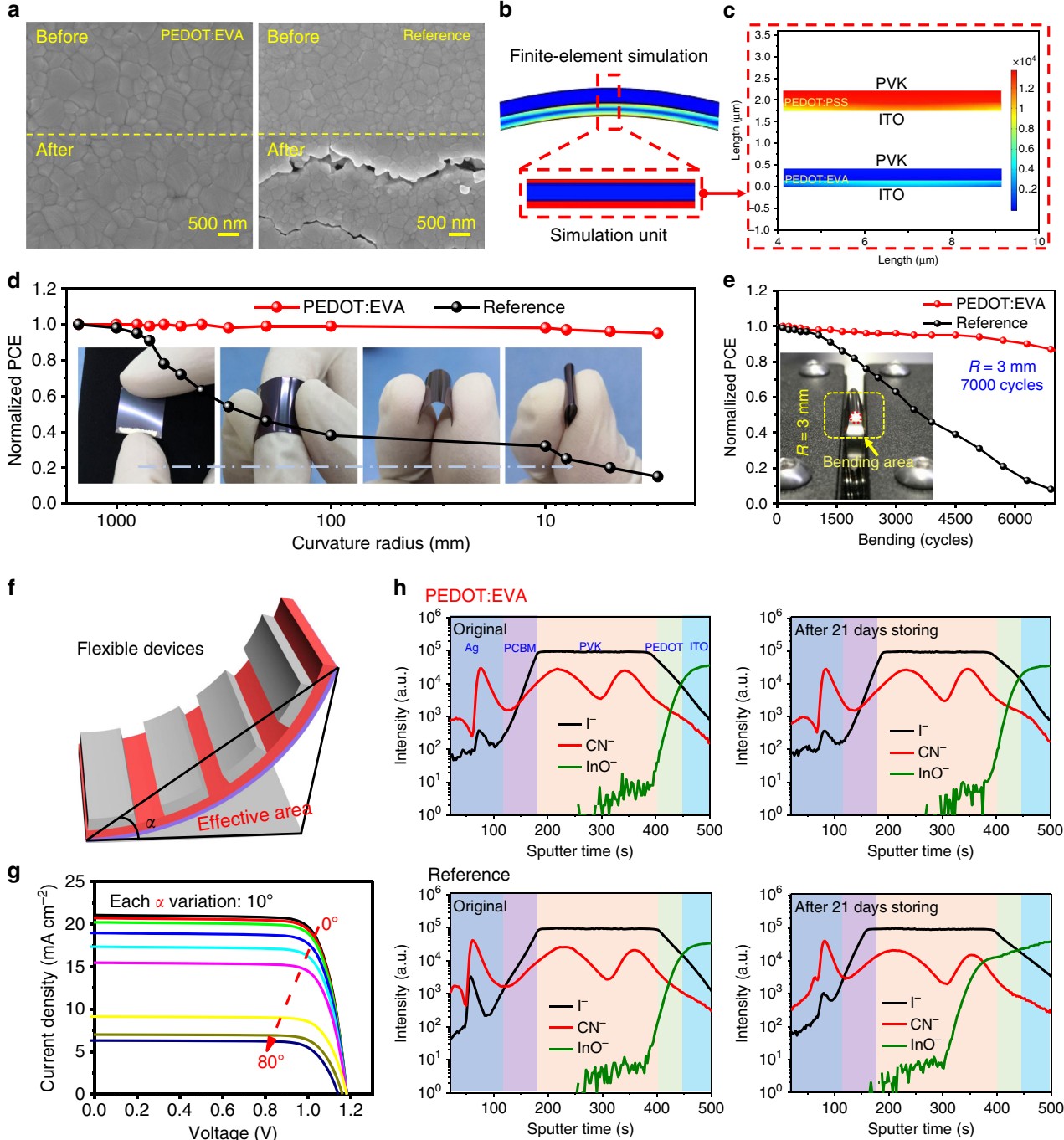

**Fig. 4 Mechanical and long-term stabilities of flexible PSCs. a** SEM images for the perovskite films before and after bending. **b**, **c** Finite-elements simulation of flexible PSCs upon PEDOT:EVA and PEDOT:PSS. **d** Normalized averaged PCE value for the flexible PSCs after bending 500 cycles with different bending radius. **e** Normalized averaged PCE value for the flexible PSCs as a function of bending cycles with radius of 3 mm. **f** The schematic diagram of device performance correction for different bending angles. **g** The PCEs of flexible devices at different bending angles. **h** Tof-SIMS elemental depth profiles for the PSCs upon PEDOT:EVA and PEDOT:PSS layers as original prepared and after 21 days, respectively.

isopropanol (99.5%, Sigma-Aldrich), rigid and flexible indium tin oxide (ITO) substrates.

**Fabrication of HTL on rigid and flexible substrate**. ITO/Glass and ITO/PET are cleaned by ultrasound in acetone, abstergent aqueous, deionized water, and iso-propyl alcohol for 45 min, respectively, then blow-dried with nitrogen (N$_2$) and treated with air plasma for 15 min. For the rigid and flexible PSCs, the PEDOT: EVA or PEDOT:PSS films with a thickness about 30 nm are printed by meniscus-coating (the blading speed is 10 mm s$^{-1}$ and the distance between scraper and ITO/glass is 50 μm). The PEDOT:PSS film is heat-annealed at 120 °C in air for 15 min. For the flexible PSCs, we use the same process as above, but the annealing

condition for the coating PEDOT:EVA or PEDOT:PSS is changed to 100 °C for 15 min. And it is worth noting that the key to fabricate efficient flexible PSCs is to ensure the flatness of the flexible substrate during the device fabrication process.

**Fabrication of perovskite solar cells**. The one-step perovskite precursor solution is obtained by dissolving 549 mg PbI$_2$, 46 mg PbBr$_2$, 150 mg HC(NH$_2$)$_2$I, 40 mg CH$_3$NH$_3$I, 0.02 wt % PU in 500 μl anhydrous DMF and 500 μl anhydrous DMSO mixture solvent in glovebox. Then, the perovskite precursor solution is meniscus-coated on the HTL/ITO/glass or PET substrates (the blading speed is 10 mm s$^{-1}$ and the distance between scraper and HTL/ITO/glass or PET substrates is 100 μm). Next, HTL/ITO/glass or PET substrates coated with perovskite precursor are

heat-annealed at 100 °C for 15 min on a heating stage. The 20 mg ml$^{-1}$ PC61BM dissolved in anhydrous chlorobenzene and 0.5 mg ml$^{-1}$ BCP dissolved in anhydrous ethanol are meniscus-coated onto the perovskite films to form the ETLs, subsequently. Finally, argentum (Ag) metal electrodes (the thickness is 90 nm for small-area devices and 120 nm for the perovskite modules) are deposited by vacuum evaporation with a vacuum of $6 \times 10^{-4}$ Torr. The preparation process of the test samples is the same as that of the device preparation before evaporation.

**Fabrication of perovskite solar modules**. The preparation condition of PSMs is exactly the same as that of the small-area optoelectronic devices, but the effective area is different. There are four sub-cells on a $60 \times 60$ mm$^2$ ITO/PET substrate and the specific dimensions of one sub-cell are 60 mm in length and 13 mm in width, respectively. In addition, the scribing techniques are applied via the rectification unit to ensure that the dislocation of each layer for the perovskite solar modules is about 0.1 mm.

**Film characterizations**. The atomic force microscope (AFM, nanoscope multi-mode Bruker) and scanning electron microscope (SEM, JEOL, JSM-7500F, Japan) at an accelerating voltage of 5.0 kV are applied to detect the surface micro-morphology of HTL or perovskite films. The ultraviolet-visible spectra (UV–vis, SHIMADZU, UV-2600 spectrophotometer) is used to measure the absorption or transmittance properties for the HTL or perovskite layers. The steady-state and lifetime spectrometer (FLS920, Edinburgh Instruments Ltd.) with the pulse width of 45 ps, the repetition rate of 0.1 MHz and the excitation fluence of 4 nJ cm$^{-2}$ from a 405 pulsed laser (the wavelength is 405 ± 8 nm) are applied to record the steady-state photoluminescence (PL) and time-resolved photoluminescence (TRPL) spectrums for the perovskite films based on different substrates. The peak emission is about 770 nm and the excitation wavelength is 470 nm. The time-correlated single-photon counting technique is used to detect the PL decay data. X-ray diffraction (XRD) and 2D-XRD measurements are recorded by Bruker D8-Discover 25 diffractometer. He I (21.22 eV) radiation line from a discharge lamp with an experimental resolution of 0.15 eV is applied for the ultraviolet photo-emission spectroscopy (UPS) measurements, and a standard procedure with a −9.0 V bias was applied to the samples to perform the UPS initiation spectra for determining the photoemission of the work function. The dual-beam ToF-SIMS depth profiling characterizations are obtained on a ToF-SIMS 5 instrument (ION-TOF GmbH) in an interlaced mode. A pulsed 30 keV Bi$^+$ ion beam was used as the analysis beam with the beam current of 1.08 pA. The analysis area was $100 \times 100$ μm$^2$ that was at the center of the sputter crater of $300 \times 300$ μm$^2$. A 2 keV Cs$^+$ ion beam was used as the sputter beam with the beam current of 120 nA. The thermo-VG Scientific ESCALAB 250 photoelectron spectrometer with a mono-chromated AlKa (1486.6 eV) X-ray source are applied to obtain the X-ray photoelectron spectroscopy (XPS) measurements for the perovskite films based on different HTLs at a base pressure in the XPS analysis chamber of $2 \times 10^{-9}$ mbar and all the recorded peaks are corrected for electrostatic effects by setting the C−C component of the C 1 s peak to 284.8 eV. The electrical impedance spectroscopy (EIS) is characterized with Zahner electrochemical workstation.

**Solar cells characterizations**. The Keithley 2400 Sourcemeter is applied to obtain the current density-voltage (J-V) and current-voltage (I-V) curves for the PSCs or PSMs. The standard silicon solar cell is corrected from NREL and the currents are detected under the solar simulator (Enli Tech, 100 mW cm$^{-2}$, AM 1.5 G irradiation). The reverse scan range and forward scan range is from 1.3 V to 0 V and from 0 V to 1.3 V, respectively, with a delay time of 30 ms, a scan rate of 0.2 V s$^{-1}$ and the each step of 8.0 mV. The monochromatic illumination (Oriel Cornerstone 260 1/4 m monochromator equipped with Oriel 70613NS QTH lamp) is used for recording the incident photo-to-electron conversion efficiency spectra (IPCE) measurements for the PSCs based on different HTLs and the calibration of the incident light is realized by the monocrystalline silicon diode. All the measurements are performed under nitrogen at room temperature. The area of PSCs is corrected by calibrated apertures (1.01 and 36.00 cm$^2$). The repeated bending cycle tests are performed by a custom-made stretching machine which is actuated by a stepper motor (Beijing Zhongke J&M). All the results of bending measurement are averaged from over 50 samples.

**Reporting summary**. Further information on research design is available in the Nature Research Reporting Summary linked to this article.

## Data availability
The data that support the findings of this study are available from the corresponding author upon reasonable request. Source data of Figs. 2c–f, 3a, c–h, 4d, e, g, h, Table 1, Supplementary Figs. 3, 5, 9, 10-20, 22, 23, 25, 27-29, 31-35, 38, 41 and Supplementary Tables 1–10 are provided as a Source Data file.

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

## Acknowledgements

Y.W.C., T.H., and X.L. thanks for support from the National Natural Science Foundation of China (NSFC) (51673091, 51803085, 21905043, 51973032 and 51833004), the National Science Fund for Distinguished Young Scholars (51425304) and NSFC-Guangdong Joint funding, China (No. U1801256). Y.S. and M.S. thanks for support from the National Key R&D Program of China (Grant Nos. 2018YFA0208501), the National Natural Science Foundation of China (Grant Nos. 51803217 and 51773206), K. C. Wong Education Foundation.

## Author contributions

X.M., Z.C., and Y.Z. contributed equally to this work. Y.W.C., Y.S., and X.H. conceived and designed the experiments. X.M. and X.H. fabricated the PSCs devices. X.M., L.Z., and X.H. completed the writing of the manuscript. Z.C. and M.S. calculated the finite-element simulation. Y.Z. and F.W. analyzed and tested the ion diffusion phenomenon. X.M., Z.X. and Z.D.H. characterized and analyzed the crystallization kinetics and morphology of PSCs. X.M. and Z.Q.H. fabricated the flexible PSCs and PSMs devices. Y.J.C., M.S., and X.L. completed the first-principles calculation. X.M., T.H., and Z.X. characterized the various photoelectric properties.

## Competing interests

The authors declare no competing interests.
