## [Peer Review File · Nature Communications]

Reviewers' comments:

Reviewer #1 (Remarks to the Author):

In this paper, the authors reported the PEDOT:EVA as novel HTL materials for flexible perovskite solar modules. The PEDOT:EVA can improve the performance of flexible perovskite solar cells due to controlling the nucleation/crystallization rate of perovskite film and passivation effect. The topic of scalable and flexible perovskite solar modules is very important in perovskite solar cell society. This work is well worth to be accepted for publishing, but there are serious scientific issues that need to be revised and improved.

- 1) All figure captions are confused to the reader. Please make more detail in all figure captions.
- 2) IPCE should provide the integrated current in Figure 3e.
- 3) Why IPCE exhibits the shoulder at the wavelength range from 700~750 nm. In addition, IPCE of the flexible cell is similar with rigid based cell. It is very strange because the transmittance of ITO/PET and ITO/glass is different.
- 4) The bandgap and transmittance of PEDOT:EVE should be provided.
- 5) The author claimed that the superior bending performance of the flexible cell is due to the excellent mechanical properties of PEDOT:EVA. However, origin of deterioration for flexible perovskite solar cell is low fatigue resistance of ITO substrate. [Energy. Environ. Sci. 8(3) 916, 2015; ACS appl. Mater. Interface 10 (5) 4697, 2018 etc.] So, I could not understand why PEDOT:EVA based flexible cell shows high mechanical stability when it bent at high bending curvature of ~3mm. In addition, Young's modulus and Poisson's ratio of PEDOT:EVA is lower than that of PEDOT:PSS. Generally, high Poisson's ratio and Young's modulus give bending properties.
- 6) In Supplementary Tab. 7, Young's modulus of PEDOT:PSS should be corrected.
- 7) Not Young's modules, it should be Young's Modulus in the figure and manuscript.

Reviewer #2 (Remarks to the Author):

The authors demonstrated the PEDOT: EVA interfacial layer between the ITO and perovskite layers facilitates the oriented crystallization of perovskite on flexible substrate and sticks the device compactly to improve the flexibility. The resultant flexible PSCs achieved the efficiencies of 19.84% and 17.55% at 1.01 cm² and 31.20 cm², respectively. However, there are some unclear contents in the current manuscript. The work needs major revision before the reconsideration of publication in Nature Communications. The work shall also be considered to publish in more special journal.

1. The manuscript published high impact journal of Nature Communication should demonstrate scientific principles rather than hype gimmicks. The work currently do not clearly show any new/shape scientific novelty. In addition, what are the differences between this work and other recent on on interface engineering related perovskite? (e.g. Nano Lett. 2019, 19, 2, 684, Adv. Mater. 2019, 31, 1902902...etc.)
2. The authors demonstrated that the PEDOT: EVA layer control the nucleation sites and crystal orientation growth. The statement does not carry important message. How does the bionic interface layer impact the nucleation and orientation growth? Why does the shorter nucleation time provide more time for grain growth to ensure the quality of perovskite film?
3. In Fig. 2f, the authors should give a whole TRPL rather than just a part. It seems that there are some problems in Fig.2f. There is a concern to get such short TRPL (9.5 vs. 16.3ns) through fitting the data in Fig.2f.
4. The interaction between Pb and carboxyl group has been reported in previous work, such as J. Am. Chem. Soc. 2019, 141, 14, 5781.
5. The authors demonstrated the EQE in the manuscript, but Fig. 3e and Fig. S13 are IPCE. Moreover,

most value in IPCE (EQE) (Fig. 3e) is lower than 85%, the integrated current from IPCE (EQE) is far lower than the J_{sc} . The authors must check all their data.

6. Apart from reported high efficiency of the present work, there is no clear scientific novelty in the work. The high efficiency should be certified. The authors also should reveal something meaningful, not hype concepts.

Reviewer #3 (Remarks to the Author):

The authors of this manuscript prepared PEDOT:EVA suspension, and used it as an hole-transporting layer in flexible perovskite solar cells. They demonstrated that their PEDOT:EVA afford perpendicular crystal growth and enhanced flexibility of the whole device, and they also achieved a power conversion efficiency up to 19.84% for 1.01 cm² flexible device. The fabrication of flexible device is an important issue in perovskite solar cells and the observation in this manuscript is interesting. Therefore, I think this manuscript can be accepted for publication after major revision as indicated in the following:

1. Please provide the basic information of PEDOT:EVA films (morphology, transmittance, electrical conductivity, and etc.) These data would help readers to understand the characteristics of PEDOT:EVA.
2. In this study, PEDOT:EVA showed the oriented crystal growth of perovskite layer. Please provide the reason why the different crystal orientation of perovskite were observed depending on different hole-transporting layers.
3. PEDOT:EVA was found to have a higher work function than PEDOT:PSS. The detailed explanation about different energy levels between PEDOT:PSS and PEDOT:EVA seems to be required.
4. It has been reported that one of the main reason of limited flexibility in flexible perovskite solar cell is the limited flexibility of ITO electrode. Thus, the detailed study about the deformation of ITO electrode depending on different interlayer materials is suggested to be required in order to understand the mechanical properties of whole devices.
5. In this study, benzene methyl sulfonic acid was used for synthesis of PEDOT, and then PEDOT:EVA suspension was prepared by miniemulsion method. What is the state of PEDOT (doped or non-doped)? Compared with the direct blending PEDOT:PSS+EVA polymer, which points are the advantage of your approach? And, if it is possible, could you provide the device performance or characteristics of PEDOT:EVA depending on different preparation conditions (ex: PEDOT/sulfonic acid ratio or PEDOT:EVA ratio or peroxide/EDOT ratio or molecular weight of EVA or etc.).

Reviewer #4 (Remarks to the Author):

This paper demonstrates a large-area flexible perovskite solar cell (PSC) with PEDOT:EVA (Poly(3,4-ethylenedioxythiophene):poly(ethylene-co-vinyl acetate)) which acts as hole transporting and soft interfacial layers inspired by the biological crystallization and flexible structure of vertebrae. The PEDOT:EVA layer which is located between brittle ITO and perovskite films could improve the mechanical flexibility of PSCs through effectively absorbing and releasing the stress, like the cartilage between vertebrae. Furthermore, the high surface energy and hydrophobicity of PEDOT:EVA affords remarkable crystallization quality and enhanced stability of the device, respectively. In details, the binding energy of Pb atoms with carboxyl group is stronger than that of sulfonic radical and the

number of the carboxyl group on PEDOT:EVA is less than that of sulfonic radical on PEDOT:PSS, resulting in fewer nucleating sites and larger grain size. The hydrophobicity of PEDOT:EVA can protect PSCs from the erosion by ion diffusion. Therefore, the PSCs based on PEDOT:EVA show the higher performance and better environmental stability than the typical PSCs based on PEDOT:PSS. Authors systematically investigated the effect of PEDOT:EVA in the interface of perovskite and ITO and the opto-electrical properties of PSCs with experimental and simulated studies. I suggest the acceptance of this manuscript in Nature Communications after addressing the following minor comments.

1. Authors suggested a PEDOT:EVA interlayer between brittle ITO and perovskite films for the improved mechanical flexibility of PSCs. Although this approach can be a good solution for the flexible PSCs based on ITO electrodes, there are other recent approaches for the enhanced flexibility and performances based on flexible silver nanowire or carbon nanotube electrodes (Kang et al, J. Mater. Chem. A 2019, 7, 1107; Jeon et al, Adv. Energy. Mater. 2019, 9, 1901204). Authors are recommended to compare merits of their approach with those other approaches.

2. Authors synthesized the PEDOT:EVA and compared the properties with PEDOT:PSS in terms of modulus, pH values and so on. However, the electrical conductivity values of PEDOT:EVA film was not described. I wonder the electrical conductivity of PEDOT:EVA film compared to PEDOT:PSS.

3. In flexible PSCs, the thickness of perovskite film critically affects the device performance and mechanical stability. I wonder the film thickness was optimized or not in this paper. If it is not, I recommend the authors show the studies about the performance and flexibility depending on the film thickness.

4. Authors explained the binding energy of Pb atom to carboxyl group is much stronger than that of sulfonic radical. Authors should add proper references for the better understanding.

5. Authors are recommended to add a comparison table in terms of the device performance and mechanical flexibility compared to other works to highlight the effect of PEDOT:EVA in the interface of perovskite and ITO.

Reviewer #5 (Remarks to the Author):

The authors have fabricated perovskite solar cells with PEDOT:EVA and achieved 19.84% PCEs on 1.01 cm² flexible PSCs with excellent mechanical stability. Furthermore, the authors measured several tool for comparison between PEDOT:PSS and PEDOT:EVA such as 2D-XRD, PL, TCSPC, XRD etc.,. However, there is no direct evidence of EVA as act like glue between ITO and perovskite. Generally, ITO is a brittle mineral that breaks when bent. But if the PEDOT:EVA acts as the glue in ITO, the property of ITO (conductivity or work function) can be changed. But there is no mentioned of these results as well as perovskite. In addition, from the DFT calculation, the authors calculated only comparison between sulfonic radical and carboxylic group. This means that any carboxylic group with PEDOT can be acts as PEDOT:EVA materials? I believe that conclusion of this paper is not clear. Finally, the authors have emphasized that PEDOT:EVA acts as “vertebrae” but there is no “why”. The exact reason for how PEDOT:EVA does not damage the device and recovers again is missing. Therefore, I give rejection this paper to nature communication.

Response to reviewers' comments

Reviewer #1

In this paper, the authors reported the PEDOT:EVA as novel HTL materials for flexible perovskite solar modules. The PEDOT:EVA can improve the performance of flexible perovskite solar cells due to controlling the nucleation/crystallization rate of perovskite film and passivation effect. The topic of scalable and flexible perovskite solar modules is very important in perovskite solar cell society. This work is well worth to be accepted for publishing, but there are serious scientific issues that need to be revised and improved.

1. Comments: All figure captions are confused to the reader. Please make more detail in all figure captions.

Reply: Thanks very much for your suggestion. We have updated the all figure captions and added more detailed instructions to make them easy to read. Revised figures are shown in the revised manuscript.

2. Comments: IPCE should provide the integrated current in Figure 3e.

Reply: Thanks for this comprehensive suggestion. We have added the integral current curve to the IPCE spectrums for the rigid and flexible devices, the corresponding descriptions and measurement results have been added to the revised manuscript and Supplementary Information. In order to facilitate the reviewers' reading, the detailed adjustment is shown in the following figures.

Fig. 3e The IPCE spectra for the flexible PSCs based on PEDOT:EVA and PEDOT:PSS substrate (reference).

Supplementary Fig.14 The IPCE spectra for the rigid PSCs based on PEDOT:EVA and reference.

3. Comments: Why IPCE exhibits the shoulder at the wavelength range from 700~750 nm. In addition, IPCE of the flexible cell is similar with rigid based cell. It is very strange because the transmittance of ITO/PET and ITO/glass is different.

Reply: Thank you very much for your reminding. In this work, the transparent electrode material selected to prepare the flexible devices is the ITO/PET substrate with a thickness of 0.125 mm. We have supplemented the intrinsic properties of PET/ITO/PEDOT:EVA and glass/ITO/PEDOT:EVA films, as shown in Supplementary Note 6. Meanwhile, for the flexible and rigid perovskite solar cells, we have revised the corresponding IPCE data description. And all the IPCE curves have been retested, and the revised data has been added to the manuscript and the corresponding integration current has also been provided. The IPCE data is really important to explain the changes in perovskite device performance and we have carefully checked and revised the description and data in the manuscript. Meanwhile, the detailed description and figure supplement are shown below. Thank you very much for your suggestions.

Supplementary Note 6. *The intrinsic properties of PEDOT:EVA films.*

The optical transmittance of PET/ITO/PEDOT:EVA or PEDOT:PSS (the thickness of PET/ITO transparent electrode is 0.125 mm) and glass/ITO/PEDOT:EVA or PEDOT:PSS is shown in Supplementary Fig. 38. Benefiting by the ultra-thin transparent electrode materials, the difference in the final transmittance for different substrate materials is not obvious, which means that the substrate materials have a little effect on the light absorption of perovskite films. The conductivity of PEDOT:EVA and ITO/PEDOT:EVA is then characterized, and the results are shown in Supplementary Tab. 9. It can be found that the conductivity of PEDOT:EVA film is better than PEDOT:PSS, and due to the excellent conductivity of ITO electrode, the conductivity of ITO/PEDOT:EVA is not significantly different from ITO/PEDOT:PSS, proving that the influence of different substrate materials on the performance of the PSCs is limited. Finally, the PEDOT:EVA and PEDOT:PSS films are morphologic characterized by AFM, as shown in Supplementary Fig. 37. Different from the typical morphology of PEDOT:PSS, the PEDOT:EVA films appear an obvious fibrous structure, which is related to the significant reduction of PSS content on the film surface. In general, the overall performance of PEDOT:EVA film is better than PEDOT:PSS, which is conducive to the improvement of device performance. Meanwhile, more importantly, the optimization of perovskite crystal quality caused by PEDOT:EVA films is also not negligible.

Supplementary Fig. 37. Atomic force microscope (AFM) images of ITO/PEDOT and ITO/PEDOT:EVA films.

Supplementary Fig. 38. Transmission spectra of glass/ITO/PEDOT:EVA, glass/ITO/PEDOT:PSS, PET/ITO/PEDOT:EVA and PET/ITO/PEDOT:PSS.

Supplementary Tab. 9. The average sheet resistance of PEDOT:EVA and the transparent electrodes coated with PEDOT:EVA.

Samples	Sheet resistance (ohm/sq)
PEDOT:EVA	43.23 ± 7.43
PEDOT:PSS	-
Glass/ITO	8.45 ± 0.76
Glass/ITO/PEDOT:PSS	463.43 ± 12.23
Glass/ITO/PEDOT:EVA	12.33 ± 1.29
PET/ITO	13.82 ± 1.22
PET/ITO/PEDOT:PSS	682.12 ± 18.23
PET/ITO/PEDOT:EVA	18.47 ± 1.45

4. Comments: The bandgap and transmittance of PEDOT:EVA should be provided.

Reply: Thanks for this comprehensive suggestion. The bandgap of PEDOT:EVA was measured by spectroelectrochemistry, which showed a 1.56 eV ($\lambda_{\text{onset}}=795\text{nm}$). And the optical transmittance curve of PEDOT:EVA film has been supplemented in the manuscript (the details are shown in the reply to comment 3). The corresponding measurement results and descriptions have been added to the manuscript.

5. Comments: The author claimed that the superior bending performance of the flexible cell is due to the excellent mechanical properties of PEDOT:EVA. However, origin of deterioration for flexible perovskite solar cell is low fatigue resistance of ITO substrate. [Energy. Environ. Sci. 8(3) 916, 2015; ACS appl. Mater. Interface 10 (5) 4697, 2018 etc.] So, I could not understand why PEDOT:EVA based flexible cell shows high mechanical stability when it bent at high bending curvature of $\sim 3\text{mm}$. In addition, Young's modulus and Poisson's ratio of PEDOT:EVA is lower than that of PEDOT:PSS. Generally, high Poisson's ratio and Young's modulus give bending properties.

Reply: Thanks for this comprehensive suggestion. We have detected the effect of PEDOT:EVA material on the ITO films (Supplementary Fig. 24, 25). The flexible transparent electrode materials

for the preparation of flexible devices are ITO/PET substrate with a thickness of 0.125mm. It can be found from the SEM images that ITO substrates covered with PEDOT:EVA verify the excellent mechanical stability under various bending radius compared with those covered with PEDOT:PSS, and no obvious micron-scale cracks appear on the surface of the film, which is due to the bonding property of PEDOT:EVA material. Meanwhile, in order to prove the substantive characteristics of PEDOT:EVA cohesiveness, we conducted the bond performance measurement, and the results are also supplemented in Supplementary Fig. 23. The PET film bonded with PEDOT:EVA material showed distinguished tensile performance, which is similar to the result in Figure1b, and this is significantly better than PEDOT:PSS ink. The reviewer's comments are very conducive to a more in-depth explanation for the advantages of PEDOT:EVA ink. All the above experimental details and corresponding descriptions have been added to the manuscript or supplementary information, and these supplements are shown below for the reviewers' convenience. Thank you for your kindly suggestions.

Supplementary Fig. 23. The stress-strain curves for the PET material bonded with PEDOT:EVA and PEDOT:PSS.

Supplementary Fig. 24. The SEM images of PET/ITO/PEDOT:EVA and PET/ITO/PEDOT:PSS bent with 4500 cycles within a curvature radius from flat to 3 mm.

Supplementary Fig. 25. Averaged sheet resistance of PET/ITO/PEDOT:EVA and PET/ITO/PEDOT:PSS measured after bending 4500 cycles with a curvature radius.

6. Comments: In Supplementary Tab. 7, Young's modulus of PEDOT:PSS should be corrected.

Reply: Thanks for this comprehensive suggestion. According to the reviewers' proposals, we have modified the Young's Modulus of PEDOT:PSS films. Young's Modulus is a common parameter in engineering design for selecting the materials of mechanical parts and is also a physical quantity describing the deformation resistance of solid materials. The Young's Modulus can be regarded as an index to measure the difficulty of elastic deformation. The larger the value of Young's Modulus usually means the greater the stress causing certain elastic deformation, and the more significant rigidity of the material. As for the flexible device, too rigid films will be prone to appear more micro cracks after bending conditions, and the mechanical stability of photoelectric device will be also pessimistic. The introduction of PEDOT:EVA buffer layer not only improves the grain size for the perovskite films, but also reduces the stress accumulation for the flexible films under various bending radius. Therefore, the mechanical stability of flexible device is significantly optimized. The corresponding modifications, descriptions and literature references have been added to the manuscript and supplementary information.

[1] Beghi, M. G., Ferrari, A. C., Teo, K. B. K., Robertson, J., Bottani, C. E., Libassi, A., Tanner, B. K., Bonding and mechanical properties of ultrathin diamond-like carbon films. *Appl. Phys. Lett.* 81, 3804-3806 (2002).

[2] Tong, L., Mehregany, M., Matus, L. G., Mechanical properties of 3C silicon carbide. *Appl. Phys. Lett.* 60, 2992-2994 (1992).

[3] Chang, H. Y., Yang, S., Lee, J., Tao, L., Hwang, W. S., Jena, D., Lu, N., Akinwande, D., High-performance, highly bendable MoS₂ transistors with high-k dielectrics for flexible low-power systems. *ACS Nano* 7, 5446-5452 (2013).

[4] Hu, X., Huang, Z., Li, F., Su, M., Huang, Z., Zhao, Z., Cai, Z., Yang, X., Meng, X., Li, P., Wang, Y., Li, M., Chen, Y., Song, Y., Nacre-inspired crystallization and elastic "brick-and-mortar" structure for a wearable perovskite solar module. *Energ. Environ. Sci.* 12, 979-987 (2019).

7. Comments: Not Young's modules, it should be Young's Modulus in the figure and manuscript.

Reply: Thanks for this comprehensive suggestion. Accurate terminologies are very important to improve the quality of manuscripts, and this kind of detailed errors should not exist. We have corrected the corresponding contents in manuscripts and Supplementary information and carefully checked other terminologies to ensure that such errors are not made.

Reviewer #2:

The authors demonstrated the PEDOT: EVA interfacial layer between the ITO and perovskite layers facilitates the oriented crystallization of perovskite on flexible substrate and sticks the device compactly to improve the flexibility. The resultant flexible PSCs achieved the efficiencies of 19.84% and 17.55% at 1.01 cm² and 31.20 cm², respectively. However, there are some unclear contents in the current manuscript. The work needs major revision before the reconsideration of publication in Nature Communications. The work shall also be considered to publish in more special journal.

1. Comments: The manuscript published high impact journal of Nature Communication should demonstrate scientific principles rather than hype gimmicks. The work currently does not clearly show any new/shape scientific novelty. In addition, what are the differences between this work and other recent on interface engineering related perovskite? (e.g. Nano Lett. 2019, 19, 2, 684, Adv. Mater. 2019, 31, 1902902...etc.)

Reply: Thank you very much for your reminding. Compared with the recent works mentioned by the reviewers, we think that our work has certain scientific novelty. However, the description of previous manuscript does not well illustrate these characteristics, and in accordance with your suggestion, we have carefully revised the manuscript to highlight the novelty.

This work is devoted to solving the key scientific problem of large-area flexible printing for the low temperature preparation of perovskite solar cells by solution method. By using the novel PEDOT:EVA buffer layer, the flexible devices can not only ensure an excellent photoelectric conversion efficiency, but also significantly optimize the mechanical stability. More importantly, the introduction of PEDOT:EVA buffer layer and the printing process can realize the high repeatability large-area preparation of the flexible devices, which has not been reported in many studies at the same time. In addition, the deep explanation of the crystallization nucleation process and the research of the interaction between carboxyl group and Pb atom have certain guiding significance for the high quality large-area printing of perovskite solar cells. For the perovskite solar cells with mesoporous structure or metal oxide material buffer layers, the characteristics of high temperature treatment and brittle metal oxide film make them difficult to be used in flexible devices.

This work focuses on the large-area printing of flexible devices and the accurate control of the nucleation behavior during film forming process, which is obviously different from the researches mentioned by the reviewers. And we also believe that the research content of this work is novel enough for the large-area printing of flexible devices. Reviewers' suggestions are very helpful for the intuitive interpretation of the manuscript, and we have revised corresponding description to the manuscript.

2. Comments: The authors demonstrated that the PEDOT:EVA layer control the nucleation sites and crystal orientation growth. The statement does not carry important message. How does the bionic interface layer impact the nucleation and orientation growth? Why does the shorter nucleation time provide more time for grain growth to ensure the quality of perovskite film?

Reply: Thank you very much for your suggestion. The control of perovskite nucleation crystallization is a collaborative process. In terms of crystallization kinetics, the preliminary regulation of nucleation process can be realized by changing the surface energy of the substrate material. In fact, the nucleation rate is controlled by a critical free energy (ΔG_c), which represents the free energy required for nuclei to be stable without being dissolved in the solution. And the Δ

G_c is also affected by variety of factors such as surface energy, molar volume, and the supersaturation of solution. For the buffer layers, larger surface energy means the greater gibbs free energy (ΔG_c), lower crystallization rate and reduces nucleation site, which can be derived from the equation (1) and (2) in Supplementary Note 4. The Lamer curves can directly reflect the nucleation process for the perovskite ink (Supplementary Note 5). The reduction of nucleation rate usually leads to shorter nucleation time, which provides longer crystallization growth time under the same VASP and annealing conditions, indicating the better and denser grain size. Meanwhile, it can be found from the results of CPMD simulation that there is a strong interaction between carboxyl group and Pb^{2+} . Compared with PSS, EVA content involved in PEDOT:EVA ink synthesis is lower, which means that PEDOT:EVA buffer layer has fewer heterogeneous nucleation sites on the surface, this will be also conducive to the improvement of the perovskite film quality. In addition, the morphology characterization results of SEM and AFM images further prove the rationality of this research method. The reviewers' questions are very helpful for the explanation of the manuscript. We have revised and supplemented the corresponding contents in the manuscript and Supplementary Note 4 and Note 5 to make them more convenient for reading and understanding.

3. Comments: In Fig. 2f, the authors should give a whole TRPL rather than just a part. It seems that there are some problems in Fig.2f. There is a concern to get such short TRPL (9.5 vs. 16.3ns) through fitting the data in Fig.2f.

Reply: Thank you very much for your reminding. The TRPL measurement is based on the ITO/PEDOT:EVA/perovskite and ITO/PEDOT:PSS/perovskite film, we have re-checked and re-tested the TRPL data of the above film samples, and the detailed results have been added in the manuscript and Supplementary Tab. 2, at the same time, the fluorescence lifetime are 9.8 and 14.6 ns respectively for the PEDOT:EVA-based film and PEDOT:PSS-based film, and the whole fluorescence lifetime and variation tendency of PEDOT:EVA-based film demonstrate a more excellent film performance. The corresponding data and description have been supplemented to the Fig 2f. The specific information is shown below for the reviewers' convenience. Thank you very much for your advice.

Fig. 2f. Time-resolved photoluminescence decay for the perovskite film based on PEDOT:EVA and reference.

Supplementary Tab.2. The carrier recombination lifetime.

HTLs	Fast phase lifetime (τ_1)	Slow phase lifetime (τ_2)
PEDOT:EVA	9.8 ns	544.1 ns
PEDOT:PSS	14.6 ns	994.5 ns

4. Comments: The interaction between Pb and carboxyl group has been reported in previous work, such as J. Am. Chem. Soc. 2019, 141, 14, 5781.

Reply: The interaction between Pb and carboxyl group is only a part of the crystal nucleation regulation. The explanation and simulation of this interaction in the manuscript is to verify that the EVA on the surface of PEDOT:EVA can provide heterogeneous nucleation sites for the crystallization of perovskite precursor solution. Most importantly, compared with PSS, less EVA content can effectively control the number of nucleation sites, providing a basis for higher quality perovskite films. This work is not only to prove the existence of this phenomenon or passivation mechanism, but mainly to confirm the advantages of PEDOT:EVA buffer layer on controlling the nucleation process of perovskite precursor solution in conjunction with the calculation and simulation results of crystal dynamics. These can provide a reasonable explanation for the optimization of perovskite film quality. Therefore, we believe that although this interaction has been reported in previous work, the collaborative research idea is still of certain scientific significance for the further research of perovskite solar cells. Thank you very much for your suggestion, which is very helpful to improve the quality of the manuscript.

5. Comments: The authors demonstrated the EQE in the manuscript, but Fig. 3e and Fig. S13 are IPCE. Moreover, most value in IPCE (EQE) (Fig. 3e) is lower than 85%, the integrated current from IPCE (EQE) is far lower than the Jsc. The authors must check all their data.

Reply: Thank you very much for your reminding. We are very sorry for this serious mistake. For the flexible and rigid perovskite solar cells, we have revised the corresponding IPCE data description. Meanwhile, all the IPCE curves have been retested, and the revised data has been added to the manuscript and the corresponding integration current has been provided. The IPCE data is really important to explain the changes in perovskite device performance. In order to facilitate the reviewers' reading, the detailed adjustment is shown in the following figures. Thank you very much for your suggestion.

Fig. 3e The IPCE spectra for the flexible PSCs based on PEDOT:EVA and PEDOT:PSS substrate (reference).

Supplementary Fig.14 The IPCE spectra for the rigid PSCs based on PEDOT:EVA and reference.

6. Comments: Apart from reported high efficiency of the present work, there is no clear scientific novelty in the work. The high efficiency should be certified. The authors also should reveal something meaningful, not hype concepts.

Reply: Thank you very much for your reminding. We have certified the PCE for flexible perovskite solar cells based on the PEDOT:EVA buffer layer, and the corresponding contents have been added to the revised manuscript or Supplementary Information (Fig. S21). Actually, during the submission process, we have certified the manuscript content for optimal large-area device efficiency. At the same time, the manuscript has been carefully examined and revised, and some of the descriptions have been added with more detailed explanations and instructions to make them easy to understand and read.

中国计量科学研究院		中国计量科学研究院	
测试报告		测试结果	
Test Report		Calibration Results	
客户名称 Client	中国科学院化学研究所 Institute of Chemistry, Chinese Academy of Sciences	有效面积 (mm ²)	短路电流 I _{sc} (A)
器具名称 Instrument	可穿戴太阳能电池模组 Wearable Solar Cell Module (PVC)	开路电压 V _{oc} (V)	最大功率 P _{max} (W)
型号规格 Type/Model	/	3602.726	0.17
出厂编号 Serial No.	28	0.17	4.72
生产厂家 Manufacturer	中国科学院化学研究所 Institute of Chemistry, Chinese Academy of Sciences	最大光生电流 I _{sc} (A)	最大功率电压 V _{max} (V)
客户地址 Address	北京非海院区中关村北一街2号 Zhongnancun North First Street 2, Beijing, P.R. China	0.15	3.60
测试日期 Date of Test	2020-01-10	填充因子 FF (%)	转换效率 (PCE) η (%)
批准人 Approved by	张介俊	63.3	14.91
地址: 中国 北京 量坛东路 18 号 Address: No. 18 Bei Sun Huan Dong 1a, Beijing, P.R. China 邮编: 100029 电话: +86-10-64719874 网址: http://www.nim.ac.cn Website:		注 Note: 1. 测试所用 mask 的面积 3602.726mm² (证书编号: C10-2020-0191)。 The mask area is 3602.726mm². Certificate No.: C10-2020-0191. 2. 此数据文件被制作成当时状态有效。 The data apply only at the time of the test for the sample (not stabilized). (以下空白)	
第 1 页 共 4 页		第 4 页 共 4 页	

Supplementary Fig. 21. The device efficiency certification report for perovskite solar cell module with a 36 cm² effective area (by National Institute of Metrology, China).

Reviewer #3:

Comments: The authors of this manuscript prepared PEDOT:EVA suspension, and used it as an hole-transporting layer in flexible perovskite solar cells. They demonstrated that their PEDOT:EVA afford perpendicular crystal growth and enhanced flexibility of the whole device, and they also achieved a power conversion efficiency up to 19.84% for 1.01 cm² flexible device. The fabrication of flexible device is an important issue in perovskite solar cells and the observation in this manuscript is interesting. Therefore, I think this manuscript can be accepted for publication after major revision as indicated in the following:

1. Comments: Please provide the basic information of PEDOT:EVA films (morphology, transmittance, electrical conductivity, and etc.) These data would help readers to understand the characteristics of PEDOT:EVA.

Reply: Thank you very much for your suggestion. The corresponding characterizations (morphology, transmittance, electrical conductivity, and etc.) and descriptions of PEDOT:EVA buffer layer have been added to the revised manuscript and Supplementary Information (Supplementary Note 6). These supplementary contents are very important for the interpretation and explanation of the manuscript. And the detailed description and figure supplement are shown below. Thank you very much for your suggestion.

Supplementary Note 6. *The intrinsic properties of PEDOT:EVA films.*

The optical transmittance of PET/ITO/PEDOT:EVA or PEDOT:PSS (the thickness of PET/ITO transparent electrode is 0.125 mm) and glass/ITO/PEDOT:EVA or PEDOT:PSS is shown in Supplementary Fig. 38. Benefiting by the ultra-thin transparent electrode materials, the difference in the final transmittance for different substrate materials is not obvious, which means that the substrate materials have a little effect on the light absorption of perovskite films. The conductivity of PEDOT:EVA and ITO/PEDOT:EVA is then characterized, and the results are shown in Supplementary Tab. 9. It can be found that the conductivity of PEDOT:EVA film is better than PEDOT:PSS, and due to the excellent conductivity of ITO electrode, the conductivity of ITO/PEDOT:EVA is not significantly different from ITO/PEDOT:PSS, proving that the influence of different substrate materials on the performance of the PSCs is limited. Finally, the PEDOT:EVA and PEDOT:PSS films are morphologic characterized by AFM, as shown in Supplementary Fig. 37. Different from the typical morphology of PEDOT:PSS, the PEDOT:EVA films appear an obvious fibrous structure, which is related to the significant reduction of PSS content on the film surface. In general, the overall performance of PEDOT:EVA film is better than PEDOT:PSS, which is conducive to the improvement of device performance. Meanwhile, more importantly, the optimization of perovskite crystal quality caused by PEDOT:EVA films is also not negligible.

Supplementary Fig. 37. Atomic force microscope (AFM) images of ITO/PEDOT and ITO/PEDOT:EVA films.

Supplementary Fig. 38. Transmission spectra of glass/ITO/PEDOT:EVA, glass/ITO/PEDOT:PSS, PET/ITO/PEDOT:EVA and PET/ITO/PEDOT:PSS.

Supplementary Tab. 9. The average sheet resistance of PEDOT:EVA and the transparent electrodes coated with PEDOT:EVA.

Samples	Sheet resistance (ohm/sq)
PEDOT:EVA	43.23 ± 7.43
PEDOT:PSS	-
Glass/ITO	8.45 ± 0.76
Glass/ITO/PEDOT:PSS	463.43 ± 12.23
Glass/ITO/PEDOT:EVA	12.33 ± 1.29
PET/ITO	13.82 ± 1.22
PET/ITO/PEDOT:PSS	682.12 ± 18.23
PET/ITO/PEDOT:EVA	18.47 ± 1.45

2. Comments: In this study, PEDOT:EVA showed the oriented crystal growth of perovskite layer. Please provide the reason why the different crystal orientation of perovskite were observed depending on different hole-transporting layers.

Reply: Thank you very much for your suggestion. Actually, the regulation of perovskite nucleation crystallization is a collaborative process and the buffer layer material is only a part of the perovskite quality. The essence of this strategy is to achieve the preliminary regulation of perovskite nucleation process by adjusting the surface energy of buffer layer material. Generally, the nucleation rate is controlled by the critical free energy (ΔG_c), which represents the free energy required for nuclei to be stable without being dissolved in the solution. And the ΔG_c is also affected by variety of factors such as surface energy, molar volume, and the supersaturation of solution. For the buffer layers, larger surface energy means the greater gibbs free energy (ΔG_c), lower crystallization rate and reduces nucleation site, which can be derived from the equation (1) and (2) in Supplementary Note 4. The Lamer curves can directly reflect the nucleation process for

the perovskite ink (Supplementary Note 5). The reduction of nucleation rate usually leads to shorter nucleation time, which provides longer crystallization growth time under the same VASP and annealing conditions, indicating the better and denser grain size. Meanwhile, it can also be found from the results of CPMD simulation that there is a strong interaction between carboxyl group and Pb atom. Compared with PSS, EVA content involved in PEDOT:EVA ink synthesis is lower, which means that PEDOT:EVA buffer layer has fewer heterogeneous nucleation sites on the surface, this will be conducive to the improvement of the perovskite quality. In addition, the 2D-XRD results and the morphology characterization results of SEM images further prove the rationality of this research method. The reviewers' questions are very helpful for the explanation of the manuscript. We have revised and supplemented the corresponding contents in the manuscript and Supplementary Note 4 and Note 5 to make them more convenient for reading and understanding.

3. Comments: PEDOT:EVA was found to have a higher work function than PEDOT:PSS. The detailed explanation about different energy levels between PEDOT:PSS and PEDOT:EVA seems to be required.

Reply: Thank you very much for your suggestion. The PEDOT:EVA film shows a higher work function than that of PEDOT:PSS, which is due to the incorporation of insulating EVA content. The moderate insulating counterpart has been verified an effective method to improve the work function of PEDOT (*Adv. Energy Mater.*, 2017, 7, 1601193; *Adv. Mater.* 2017, 29, 1703236.). The corresponding explanation has been revised in the manuscript.

4. Comments: It has been reported that one of the main reason of limited flexibility in flexible perovskite solar cell is the limited flexibility of ITO electrode. Thus, the detailed study about the deformation of ITO electrode depending on different interlayer materials is suggested to be required in order to understand the mechanical properties of whole devices.

Reply: Thanks for this comprehensive suggestion. We have detected the effect of PEDOT:EVA material on the ITO films (Supplementary Fig. 23, 24). The flexible transparent electrodes for the preparation of flexible devices are ITO/PET substrate with a thickness of 0.125 mm. And the bending performance and morphology characterization of ITO/PET coated with PEDOT:EVA or PEDOT:PSS buffer layers were tested. It can be found from the SEM images that ITO substrates covered with PEDOT:EVA verify the excellent mechanical stability under various bending radius compared with those covered with PEDOT:PSS, and no obvious micron-scale cracks appear on the surface of the film, which is due to the bonding property of PEDOT:EVA material. Meanwhile, in order to prove the substantive characteristics of PEDOT:EVA cohesiveness, we conducted the bond performance measurement, and the results are also supplemented in Supplementary Note 8 and Supplementary Fig. 22. The PET film bonded with PEDOT:EVA material showed distinguished tensile performance, which is similar to the result in Figure1b, and this is significantly better than PEDOT:PSS ink. The reviewer's comments are very conducive to a more in-depth explanation for the advantages of PEDOT:EVA ink. Finally, all the above experimental details and corresponding descriptions have been added to the manuscript or supplementary information. These supplements are also shown below for the reviewers' convenience. Thank you for your kindly suggestions.

Supplementary Fig. 23. The stress-strain curves for the PET material bonded with PEDOT:EVA and PEDOT:PSS.

Supplementary Fig. 24. The SEM images of PET/ITO/PEDOT:EVA and PET/ITO/PEDOT:PSS bent with 4500 cycles within a curvature radius from flat to 3 mm.

Supplementary Fig. 25. Averaged sheet resistance of PET/ITO/PEDOT:EVA and PET/ITO/PEDOT:PSS measured after bending 4500 cycles with a curvature radius.

5. Comments: In this study, benzene methyl sulfonic acid was used for synthesis of PEDOT, and then PEDOT:EVA suspension was prepared by miniemulsion method. What is the state of PEDOT (doped or non-doped)? Compared with the direct blending PEDOT:PSS+EVA polymer, which points are the advantage of your approach? And, if it is possible, could you provide the device performance or characteristics of PEDOT:EVA depending on different preparation conditions (ex: PEDOT/sulfonic acid ratio or PEDOT: EVA ratio or peroxide/EDOT ratio or molecular weight of EVA or etc.)

Reply: Thanks for this comprehensive suggestion. From the characterization results of Raman spectrum, Dynamic light scattering and X-ray photoelectron spectroscopy, the PEDOT in PEDOT:EVA ink still exists in a state similar to PEDOT:PSS suspension, but PEDOT:EVA ink is a neutral solution, and the content of PSS on the surface is significantly reduced after film formation. Compared with the PEDOT ink directly doped with EVA solution, PEDOT:EVA suspension obtained by microemulsion polymerization is more stable and homogeneous. In fact, the EVA solution for the preparation of PEDOT:EVA ink is dissolved in chlorobenzene, and the

simple blending makes it difficult to blend well with PEDOT:PSS ink, which is usually dispersed in aqueous solution, therefore, the microemulsion method has become an optimistic approach. According to the reviewers' suggestions, we have supplemented the device performance and PEDOT:EVA buffer layer characteristics with different PEDOT/sulfonic acid ratio, PEDOT: EVA ratio, peroxide/EDOT ratio and molecular weight of EVA. These contents have been taken into account in the experiment design, but considering the length of words, we only selected the optimal preparation parameters. The corresponding description and data have been added to the manuscript and Supplementary Note 1, which are very helpful to highlight the advantages of PEDOT:EVA buffer layer, and the details are shown in the table below Thank you very much for your suggestions.

Supplementary Table. The device performance of rigid perovskite solar cells under different synthetic conditions.

Dibenzoyl peroxide (g)	3,4-ethylenedioxythiophene (g)	Benzene methyl sulfonic acid (g)	EVA (mg/ml)	PCE (%)
0.85	0.60	2.50	20	20.31
1.05	0.60	2.50	20	22.16
1.25	0.60	2.50	20	20.87
1.05	0.45	2.50	20	18.32
1.05	0.75	2.50	20	19.44
1.05	0.60	1.50	20	17.75
1.05	0.60	3.50	20	20.32
1.05	0.60	2.50	5	17.32
1.05	0.60	2.50	10	19.97
1.05	0.60	2.50	30	19.65

Reviewer #4:

Comments: This paper demonstrates a large-area flexible perovskite solar cell (PSC) with PEDOT:EVA (Poly(3,4-ethylenedioxythiophene):poly(ethylene-co-vinyl acetate)) which acts as hole transporting and soft interfacial layers inspired by the biological crystallization and flexible structure of vertebrae. The PEDOT:EVA layer which is located between brittle ITO and perovskite films could improve the mechanical flexibility of PSCs through effectively absorbing and releasing the stress, like the cartilage between vertebrae. Furthermore, the high surface energy and hydrophobicity of PEDOT:EVA affords remarkable crystallization quality and enhanced stability of the device, respectively. In details, the binding energy of Pb atoms with carboxyl group is stronger than that of sulfonic radical and the number of the carboxyl group on PEDOT:EVA is less than that of sulfonic radical on PEDOT:PSS, resulting in fewer nucleating sites and larger grain size. The hydrophobicity of PEDOT:EVA can protect PSCs from the erosion by ion diffusion. Therefore, the PSCs based on PEDOT:EVA show the higher performance and better environmental stability than the typical PSCs based on PEDOT:PSS. Authors systematically investigated the effect of PEDOT:EVA in the interface of perovskite and ITO and the opto-electrical properties of PSCs with experimental and simulated studies. I suggest the acceptance of this manuscript in Nature Communications after addressing the following minor comments.

1. Comments: Authors suggested a PEDOT:EVA interlayer between brittle ITO and perovskite films for the improved mechanical flexibility of PSCs. Although this approach can be a good solution for the flexible PSCs based on ITO electrodes, there are other recent approaches for the enhanced flexibility and performances based on flexible silver nanowire or carbon nanotube electrodes (Kang et al, J. Mater. Chem. A 2019, 7, 1107; Jeon et al, Adv. Energy. Mater. 2019, 9, 1901204). Authors are recommended to compare merits of their approach with those other approaches.

Reply: Thank you very much for your question. In the previous development of perovskite solar cells, ITO transparent electrode was usually the best electrode material to ensure device efficiency. However, due to the characteristics of brittleness, ITO electrode does not demonstrate perfect device performance in the preparation of flexible devices. Therefore, flexible transparent electrodes such as silver nanowires and carbon nanotubes have gradually become substitutes for the effective flexible devices. Although these materials develop relatively good mechanical stability, they have not made outstanding progress in device efficiency, especially in the large-area flexible devices. Meanwhile, silver nanowires are easy to fall off the substrate and the resistance at the junction of carbon nanotubes is too large, which seriously limit their further development and reproducible preparation.

This work is based on this modification and optimization for the hole buffer layer. On the one hand, ITO and PEDOT:EVA buffer layer are used to ensure excellent device efficiency, on the other hand, PEDOT layer is used to bond perovskite and ITO materials to improve the bending resistance for the flexible devices. At the same time, these phenomena are explained and analyzed by means of crystal dynamics simulation and crystallization nucleation dynamics calculation, and the large-area flexible devices are prepared to preliminarily verify their practical application in small power devices. We believe that different research strategies are aimed at achieving perfect device performance and commercializing perovskite solar cells, and the current results in this manuscript are more advantageous and feasible than the previous work. According to the reviewers' suggestions, the corresponding description and comparative advantages have been added in the manuscript, which is helpful for reading and understanding the development status of flexible

printing transparent electrode, thank you very much for your suggestion.

2. Comments: Authors synthesized the PEDOT:EVA and compared the properties with PEDOT:PSS in terms of modulus, pH values and so on. However, the electrical conductivity values of PEDOT:EVA film was not described. I wonder the electrical conductivity of PEDOT:EVA film compared to PEDOT:PSS.

Reply: Thank you very much for your suggestion. The corresponding characterizations and description about micro-morphology, optical transmittance and electrical conductivity for the PEDOT:EVA buffer layer have been supplemented in the revised manuscript and Supplementary Tab. 9. The specific contents are shown in the following table. These contents are very important for the interpretation and explanation of the manuscript.

Supplementary Tab. 9. The average sheet resistance of PEDOT:EVA and the transparent electrodes coated with PEDOT:EVA.

Samples	Sheet resistance (ohm/sq)
PEDOT:EVA	43.23 ± 7.43
PEDOT:PSS	-
Glass/ITO	8.45 ± 0.76
Glass/ITO/PEDOT:PSS	463.43 ± 12.23
Glass/ITO/PEDOT:EVA	12.33 ± 1.29
PET/ITO	13.82 ± 1.22
PET/ITO/PEDOT:PSS	682.12 ± 18.23
PET/ITO/PEDOT:EVA	18.47 ± 1.45

3. Comments: In flexible PSCs, the thickness of perovskite film critically affects the device performance and mechanical stability. I wonder the film thickness was optimized or not in this paper. If it is not, I recommend the authors show the studies about the performance and flexibility depending on the film thickness.

Reply: Thank you very much for your suggestion. Film thickness has a very important influence on device performance and mechanical stability. In Supplementary Note 2, we have conducted statistics and analysis on the device performance with different meniscus moving speeds, meniscus spacing and VASP pressure. Under such conditions, the film thickness actually becomes the medium that determines the overall performance for the perovskite solar cells, because the variation of the above parameters always leads to the different film thickness and quality. In order to facilitate understanding and reading, we have added the summary of device performance under different film thickness conditions, and the results have been added to the Supplementary Fig. 13, and also shown below. The analysis combining the printing parameters with the film thickness can more intuitively reflect the variation law of device performance, and these contents are well suited to be supplemented to the manuscript. Thank you very much for your kindly suggestions.

Supplementary Fig. 13. The effect of perovskite film thickness on device performance.

4. Comments: Authors explained the binding energy of Pb atom to carboxyl group is much stronger than that of sulfonic radical. Authors should add proper references for the better understanding.

Reply: Thanks for this comprehensive suggestion. Actually, the interaction between carboxyl group and Pb atom has indeed been reported in previous work, and we have added the corresponding references to the revised manuscript and Supplementary Information. This work is to prove the interaction between Pb atom with carboxyl group via the crystal dynamics simulation, so as to better explain the regulation of nucleation site on crystal quality, which is a supplement to the calculation of crystallization nucleation dynamics. Appropriate literature citation can more convincingly demonstrate the validity for the research content in manuscript, and the reviewers' reminding is very helpful to improve the quality of the revised manuscript.

[1] O'keeffe, P., Catone, D., Paladini, A., Toschi, F., Turchini, S., Avaldi, L., Martelli, F., Agresti, A., Pescetelli, S., Del Rio Castillo, A. E., Bonaccorso, F., Di Carlo, A., Graphene-Induced Improvements of Perovskite Solar Cell Stability: Effects on Hot-Carriers. *Nano Lett.* **19**, 684-691 (2019).

[2] Chen, J., Zhao, X., Kim, S. G., Park, N. G., Multifunctional Chemical Linker Imidazoleacetic Acid Hydrochloride for 21% Efficient and Stable Planar Perovskite Solar Cells. *Adv. Mater.* **31**, 1902902 (2019).

[3] Yang, S., Dai, J., Yu, Z., Shao, Y., Zhou, Y., Xiao, X., Zeng, X. C., Huang, J., Tailoring passivation molecular structures for extremely small open-circuit voltage loss in perovskite solar cells. *J. Am. Chem. Soc.* **141**, 5781-5787 (2019).

5. Comments: Authors are recommended to add a comparison table in terms of the device performance and mechanical flexibility compared to other works to highlight the effect of PEDOT:EVA in the interface of perovskite and ITO.

Reply: Thanks for this comprehensive suggestion. We have counted the performance of the flexible devices in this work compared with the previous work, and the results are shown in Supplementary Fig.22. In the statistical graph, the recent research results based on the perovskite layer optimization, interface layer or transparent electrode modification and device performance with different effective areas are respectively collected., and the perovskite solar cells based on PEDOT:EVA buffer layer verify great advantages in both small-area flexible devices and large-area modules. Meanwhile, according to the reviewers' suggestions, we also made statistics and

summary on the mechanical stability of the flexible device, and the results were supplemented in Supplementary Fig. 30. These supplements are conducive to reflecting the advantages of PEDOT:PSS buffer layer in the comprehensive performance for the flexible perovskite solar cells, which is very necessary, and thank you very much for your advice.

Supplementary Fig. 31. The summary of residual PCE of recently reported flexible PSCs after bending cycles.

Reviewer #5:

Comments: The authors have fabricated perovskite solar cells with PEDOT:EVA and achieved 19.84% PCEs on 1.01 cm² flexible PSCs with excellent mechanical stability. Furthermore, the authors measured several tool for comparison between PEDOT:PSS and PEDOT:EVA such as 2D-XRD, PL, TCSPC, XRD etc.,. However, there is no direct evidence of EVA as act like glue between ITO and perovskite. Generally, ITO is a brittle mineral that breaks when bent. But if the PEDOT:EVA acts as the glue in ITO, the property of ITO (conductivity or work function) can be changed. But there is no mentioned of these results as well as perovskite. In addition, from the DFT calculation, the authors calculated only comparison between sulfonic radical and carboxylic group. This means that any carboxylic group with PEDOT can be acts as PEDOT:EVA materials? I believe that conclusion of this paper is not clear. Finally, the authors have emphasized that PEDOT:EVA acts as “vertebrae” but there is no “why”. The exact reason for how PEDOT:EVA does not damage the device and recovers again is missing. Therefore, I give rejection this paper to nature communication.

Reply: Thanks for these comprehensive suggestions. We have tried our best to revise the description and supplement the corresponding data in the manuscript according to the reviewers' suggestions. And we hope that the revised manuscript has the potential to be published in Nature Communication.

First, the PEDOT:EVA is designed as a blocking glue, which bonds ITO and perovskite together, and the cohesiveness of PEDOT:EVA ink is demonstrated by the strain-stress measurement (Fig. 1b and Supplementary Fig. 23). Compared with the PET plastic bonded with PEDOT:PSS, the PET plastic bonded with PEDOT:EVA exhibits significantly superior bonding performance, which can maintain mechanical stability until fracture occurs at a strain ratio of 17~18% (for the PEDOT:PSS, the fracture-tensile ratio is no more than 10%), confirming that the PEDOT:EVA ink has an optimistic cohesiveness (*J. Am. Chem. Soc.* **2017**, *139*, 8678; *Adv. Energy Mater.* **2020**, *10*, 1903609). The cohesiveness of EVA originates from the adsorption interaction of EVA emulsion (Van Der Waals or hydrogen-bonding interaction) in the PEDOT:EVA ink prepared by the miniemulsion method, which has been reported in previous adhesive work (*J. Appl. Polym. Sci.* 2002, *83*, 719; *U. S. Patent* **3**, 661 (1972); *U. S. Patent* **3**, 448 (1969) and *U. S. Patent* **2**, 543 (1951)). Therefore, the PEDOT:EVA layer accordingly becomes the center of stress absorption and release like the spinal cartilage, improving the mechanical toughness of photoelectric devices. Then, the contribution of PEDOT:EVA and PEDOT:PSS to the bending resistance are explored through the SEM images characterization for the ITO and perovskite films respectively. For ITO transparent electrode coated with PEDOT:EVA, no obvious micron-scale crack appears after 4500 cycles, which is far better than that coated with PEDOT:PSS, and the corresponding results are also shown in Supplementary Note 9, Supplementary Fig. 24 and Supplementary Fig. 25 (the flexible transparent electrodes for the preparation of flexible perovskite solar cells are ITO/PET substrate with a thickness of 0.125 mm). In addition, the conductivity of ITO flexible electrodes covered with PEDOT:EVA and PEDOT:PSS is also measured by the four-point probe measurement, and the results are shown in Supplementary Note 6 and Supplementary Tab. 9 (the detailed optical, electrical properties and characteristics of PEDOT: EVA are shown in Supplementary Fig.32-38). And the corresponding supplements about the PEDOT:EVA material are shown below.

Supplementary Fig. 23. The stress-strain curves for the PET material bonded with PEDOT:EVA and PEDOT:PSS.

Supplementary Fig. 24. The SEM images of PET/ITO/PEDOT:EVA and PET/ITO/PEDOT:PSS bent with 4500 cycles within a curvature radius from flat to 3 mm.

Supplementary Fig. 25. Averaged sheet resistance of PET/ITO/PEDOT:EVA and PET/ITO/PEDOT:PSS measured after bending 4500 cycles with a curvature radius.

Supplementary Fig. 37. Atomic force microscope (AFM) images of ITO/PEDOT and ITO/PEDOT:EVA films.

Supplementary Fig. 38. Transmission spectra of glass/ITO/PEDOT:EVA, glass/ITO/PEDOT:PSS, PET/ITO/PEDOT:EVA and PET/ITO/PEDOT:PSS.

Supplementary Tab. 9. The average sheet resistance of PEDOT:EVA and the transparent electrodes coated with PEDOT:EVA.

Samples	Sheet resistance (ohm/sq)
PEDOT:EVA	43.23 ± 7.43
PEDOT:PSS	-
Glass/ITO	8.45 ± 0.76
Glass/ITO/PEDOT:PSS	463.43 ± 12.23
Glass/ITO/PEDOT:EVA	12.33 ± 1.29
PET/ITO	13.82 ± 1.22
PET/ITO/PEDOT:PSS	682.12 ± 18.23
PET/ITO/PEDOT:EVA	18.47 ± 1.45

Next, in the DFT simulation, the simulation objects are the interaction between perovskite with MAI or PbI_2 as the face and the PSS or EVA materials, this is not simple functional groups such as sulfonic acid or carboxyl group. Meanwhile, PEDOT material does not have these functional groups (sulfonic acid or carboxyl acid). The interaction between carboxyl group and Pb has been reported many times in the previous reports of various additive materials, confirming the effectiveness of this interaction (*J. Am. Chem. Soc.* **2019**, *141*, 5781). It is worth noting that the interaction between Pb and carboxyl group is only a part of the crystal nucleation regulation. The explanation and simulation of this interaction in the manuscript is to verify that the EVA on the surface of PEDOT:EVA can provide heterogeneous nucleation sites for the crystallization of perovskite precursor solution. More importantly, compared with PSS, less EVA content can effectively control the number of nucleation sites, providing a basis for higher quality perovskite films. This work is not only to prove the existence of this phenomenon or passivation mechanism, but mainly to confirm the advantages of PEDOT:EVA buffer layer on controlling the nucleation process of perovskite precursor solution in conjunction with the calculation and simulation results of crystal dynamics, which demonstrate a reasonable explanation for the optimization of

perovskite film quality. So, we believe that although this interaction has been reported, the collaborative research idea is still of certain scientific significance for the further research of perovskite solar cells.

Finally, the “vertebrae” structure does not refer to the PEDOT:EVA monolayer, but to the ITO/PEDOT/perovskite (brittle layer/soft adhesive layer/brittle layer) structure. For the normal operation of vertebrae structure, this structure can satisfy sufficient bending resistance, while the hydrophobic adhesive layer facilitates the formation of the dense and complete upper crystalline material. Similar work has been reported successively in biological studies (*Adv. Mater.* **2017**, *29*, 1605903; *Adv. Mater.* **2016**, *28*, 694; *Adv. Mater.* **2018**, *30*, 1704876), therefore, the connection and application of flexible perovskite solar cells in this regard is feasible and practical. In order to verify the advantages of PEDOT:EVA buffer layer in bending resistance, the mechanical stability and morphology characterization for the ITO films covered with PEDOT:EVA and meniscus-printed perovskite films are measured, and these results have confirmed the feasibility of “vertebrae” biomimetic structure. The reviewer's comments are very conducive to a more in-depth explanation for the advantages of PEDOT:EVA ink. All the above experimental details and corresponding descriptions have been added to the manuscript or supplementary information. Thank you for your kindly suggestions.

In fact, in addition to excellent device performance (the PCE of large-area PSMs devices has also been certified, as shown in the figure below), perovskite solar cells have a unique advantage of potential applications in flexible optoelectronic devices. And the high repeatability preparation based on large-area printing process for the commercial development of perovskite devices is crucial. This work is based on this research status, the flexible large-area perovskite solar cells with excellent mechanical stability and device efficiency were fabricated by using bionic structure, novel interface layer materials and the meniscus-printing technology, and the feasibility of these method in large-area modules and the practicability in low-power devices are verified and demonstrated. At the same time, this phenomenon is further explained and introduced by means of crystal dynamics simulation and crystallization nucleation dynamics calculation. We believe that both the development of large-area printing technology for the flexible perovskite solar cells and the in-depth research strategy of crystal dynamics can match the conditions for publishing in Nature Communication.

Supplementary Fig. 21. The device efficiency certification report for perovskite solar cell module with a 36 cm² effective area (by National Institute of Metrology, China).

Reviewers' comments:

Reviewer #1 (Remarks to the Author):

Revised manuscript is now satisfactory and paper can be now be published.

Reviewer #2 (Remarks to the Author):

The authors have addressed the questions accordingly. The manuscript can be accepted for publication in Nature Communication as is.

Reviewer #3 (Remarks to the Author):

The authors properly addressed all the issues raised by this reviewer. Therefore, I am not negative for the publication of this paper in Nature Communications.

Reviewer #4 (Remarks to the Author):

Authors did not properly address some comments raised by the reviewers. This manuscript should be reconsidered after addressing the following comments.

1. Comment 1 from reviewer 4: Authors answered that they added corresponding description and comparative advantages in the manuscript. However, I could not find this content in the manuscript.
2. Comment 2 from reviewer 4: Authors mentioned that variation of several parameters (meniscus moving speeds, meniscus spacing and VASP pressure) always leads to the different film thickness and quality. Then, there should be some variation in the data in Supplementary Fig. 13. Authors need to provide average and standard deviation for the data.
3. Comment 5 from reviewer 4: Authors added Supplementary Fig. 31 to address the comment about adding a comparison table. However, Supplementary Fig. 31 does not provide useful information for comparison with the other works. It does not include the corresponding references for the data points in the graph. In addition, it does not show the detailed device structure, bending radius, absolute PCE values, which will be required for the fair comparison with the other works.

Reviewer #5 (Remarks to the Author):

I believe that the author mentioned PEDOT:EVA adhesive property with PET. But my comment is that how ITO with PEDOT:EVA recovered the performance including solar cells and electrical property after bending. As the authors mentioned in revision letter, ITO is cracked after bending which mean that ITO composed In-Sn-O chemical structure is destroyed. However, the authors did not clearly mention this point. I believe that this point is very important issue for this paper and need to carefully address "why". And from the DFT calculation, the author only compared to Pb atom with carboxyl group and sulfonic radical which indicates that EVA materials is only important for carboxylic group. This conclusion can be induced to any materials with carboxylic group can replace the PSS. Therefore, I believe that the authors need more clear evidence why EVA is important with PEDOT compared to other materials. Finally, "vertebrae" means the similar issue with "recover" but the authors did not clearly mention "why"

Therefore, I don't recommend publication of this paper in nature communications.

Response to reviewers' comments

Reviewer #1:

Comments: Revised manuscript is now satisfactory and paper can be now be published.

Reply: We thank the referee for a constructive review process.

Reviewer #2:

Comments: The authors have addressed the questions accordingly. The manuscript can be accepted for publication in Nature Communication as is.

Reply: We thank the reviewer for the positive evaluation of our revised manuscript.

Reviewer #3:

Comments: The authors properly addressed all the issues raised by this reviewer. Therefore, I am not negative for the publication of this paper in Nature Communications.

Reply: We thank the referee for a constructive review process.

Reviewer #4:

Comments: Authors did not properly address some comments raised by the reviewers. This manuscript should be reconsidered after addressing the following comments.

1. Comments: Authors answered that they added corresponding description and comparative advantages in the manuscript. However, I could not find this content in the manuscript.

Reply: Thank you very much for your kindly reminding. We are very sorry for missing the corresponding content of revised manuscript. We have further revised the corresponding descriptions and added these to the revised manuscript.

The specific contents are “ITO electrode is usually the best conductive material to ensure device performance for the scalable PSCs. Unfortunately, due to the feature of brittleness, ITO electrode does not demonstrate perfect performance in flexible photoelectric devices. Although some flexible transparent electrodes such as silver nanowires and carbon nanotubes have shown the potential for application in flexible PSCs, they have not made outstanding progress in device efficiency, especially in the large-area devices³⁴⁻³⁹. Therefore, flexible PSCs based on the modified ITO electrode are also worth for further investigation.” and “ITO and PEDOT:EVA buffer layer are used to ensure excellent device efficiency, meanwhile, PEDOT:EVA layer is used to bond perovskite and ITO materials to improve the bending resistance for the flexible devices.”, which have been added to Page 2, paragraph 1, line 3 and Page 2, paragraph 2, line 8 in the revised manuscript. We also marked it in red. Thanks again for your help.

34. Kang, S., Jeong, J., Cho, S., Yoon, Y. J., Park, S., Lim, S., Kim, J. Y., Ko, H. Ultrathin, lightweight and flexible perovskite solar cells with an excellent power-per-weight performance. *J. Mater. Chem. A*, **7**, 1107-1114 (2019).
35. Kim, A., Won, Y., Woo, K., Jeong, S., Moon, J. All-Solution-Processed Indium-Free Transparent Composite Electrodes based on Ag Nanowire and Metal Oxide for Thin-Film Solar Cells. *Adv. Funct. Mater.* **24**, 2462-2471 (2014).
36. Jeon, I., Yoon, J., Kim, U., Lee, C., Xiang, R., Shawky, A., Xi, J., Byeon, J., Lee, H. M., Choi, M., Maruyama, S., Matsuo, Y. High-Performance Solution-Processed Double-Walled Carbon Nanotube Transparent Electrode for Perovskite Solar Cells. *Adv. Energy Mater.* **9**, 1901204 (2019).
37. Luo, Q., Ma, H., Zhang, Y., Yin, X., Yao, Z., Wang, N., Li, J., Fan, S., Jiang, K., Lin, H. Cross-stacked superaligned carbon nanotube electrodes for efficient hole conductor-free perovskite solar cells. *J. Mater. Chem. A*, **4**, 5569-5577 (2016).
38. Li, Y., Xu, G., Cui, C., Li, Y. Flexible and Semitransparent Organic Solar Cells. *Adv. Energy Mater.* **8**, 1701791 (2018).
39. Sannicolo, T., Lagrange, M., Cabos, A., Celle, C., Simonato, J. P., Bellet, D. Metallic Nanowire-Based Transparent Electrodes for Next Generation Flexible Devices: a Review. *Small* **12**, 6052-6075 (2016).

2. Comments: Authors mentioned that variation of several parameters (meniscus moving speeds, meniscus spacing and VASP pressure) always leads to the different film thickness and quality. Then, there should be some variation in the data in Supplementary Fig. 13. Authors need to provide average and standard deviation for the data.

Reply: Thank you very much for your comprehensive suggestion. In order to reflect clearly the specific relationship between film thickness variation with the printing parameters or device

efficiency, we have made statistics and summaries for each parameter and provided the average and standard deviations for the data. The supplement of these content is very important to guarantee the reproducibility for the printing parameters in manuscript. The reviewers' questions are very careful and professional, for which we are very grateful. The specific data are shown below and are added in Supplementary Fig. 13 and Supplementary Tab. 4. Meanwhile, we have supplemented the corresponding description in the revised manuscript (Page 6, paragraph 2, line 14).

Supplementary Fig. 13. The effect of perovskite film thickness based on different meniscus moving speed, meniscus spacing and VASP pressure for the device performance.

Supplementary Tab. 4 The effect of perovskite film thickness based on different meniscus moving speed, meniscus spacing and VASP pressure for the device performance.

Constant	Moving speed (mm/s)	Film thickness (nm)	PCE (%)
Meniscus spacing (50 μm) VASP pressure (20 Pa)	5	211 \pm 17.1	16.11 \pm 0.51
	6	233 \pm 17.7	16.52 \pm 0.28
	7	259 \pm 18.4	17.62 \pm 0.32
	8	293 \pm 16.7	18.01 \pm 0.35
	9	323 \pm 17.3	18.53 \pm 0.23
	10	355 \pm 17.5	19.51 \pm 0.21
	12	386 \pm 16.8	18.83 \pm 0.24
	14	417 \pm 17.2	18.21 \pm 0.32
	16	440 \pm 18.3	17.45 \pm 0.31
	18	475 \pm 20.1	16.76 \pm 0.27
20	512 \pm 18.6	15.42 \pm 0.42	
Constant	Meniscus spacing (μm)	Film thickness (nm)	PCE (%)
Meniscus moving speed (10 mm/s) VASP pressure (20 Pa)	10	52 \pm 5.2	5.22 \pm 1.21
	20	150 \pm 9.4	12.12 \pm 0.79
	30	244 \pm 10.3	16.21 \pm 0.85
	40	301 \pm 11.3	17.83 \pm 0.29
	50	352 \pm 10.6	19.35 \pm 0.37
	70	431 \pm 17.3	18.23 \pm 0.33
	90	492 \pm 24.5	16.65 \pm 0.44
	110	525 \pm 36.4	14.23 \pm 1.21
130	629 \pm 40.2	11.34 \pm 2.32	
Constant	Pressure (Pa)	Film thickness (nm)	PCE (%)
Meniscus spacing (50 μm)	10	324 \pm 6.7	18.51 \pm 0.34
	20	342 \pm 7.6	19.12 \pm 0.36

Meniscus moving speed (10 mm/s)	50	345 ± 6.5	18.41 ± 0.51
	80	364 ± 9.4	18.12 ± 0.44
	110	361 ± 10.5	18.01 ± 0.46
	150	367 ± 9.7	17.45 ± 0.62
	300	375 ± 11.4	17.26 ± 0.71
	500	381 ± 10.9	16.21 ± 1.22
	700	392 ± 12.6	15.23 ± 1.34
	900	394 ± 11.0	14.21 ± 1.65
	1100	401 ± 13.1	13.02 ± 2.01

3. Comments: Authors added Supplementary Fig. 31 to address the comment about adding a comparison table. However, Supplementary Fig. 31 does not provide useful information for comparison with the other works. It does not include the corresponding references for the data points in the graph. In addition, it does not show the detailed device structure, bending radius, absolute PCE values, which will be required for the fair comparison with the other works.

Reply: Thank you very much for your suggestion. We have re-summarized and re-depicted the Supplementary Fig.31. Your suggestion is very effective for a fair comparison of the manuscript content with other flexible perovskite optoelectronic reports. The revised Supplementary Fig. 31, a new Supplementary Tab. 10 and the corresponding references are shown below and in the revised Supplementary Information. The supplement of these contents more directly reflects the comprehensive level for the flexible device performance in this work. We hope that the revised content can meet the expected results of reviewers. Thanks again for your kindly help.

Supplementary Fig.31 The summary of residual PCE for the recently reported flexible PSCs after bending cycles.

Supplementary Tab.10 The summary of residual PCE for the recently reported flexible PSCs after bending cycles.

Structure	PCE (%)	Bending cycles/ bending radius	Residual PCE (%)	Journal
-----------	---------	-----------------------------------	------------------	---------

PET/ITO/E-SnO ₂ /PVK/Spiro-OMeTAD/Au	18.28	500 cycles/7 mm	16.84	Nat. Commun. ²¹
PET/ITO/SnO ₂ /PVK/C ₆₀ -SAM/PVK/ Spiro-OMeTAD/Au	17.96	340 cycles/5 mm	14.30	Nano Energy ²²
PEN/ITO/SnO ₂ /PVK/Spiro-OMeTAD/Au	19.38	500 cycles/10 mm	17.83	Adv. Funct. Mater. ²³
Cellophane/OMO/CPTA/PVK/ Spiro-OMeTAD/Au	13.00	1000 cycles/1 mm	12.45	Sol. Energy ²⁴
PET/ITO/SnO ₂ /C ₆₀ -SAM/PVK/ Spiro-OMeTAD/Au	17.43	1000 cycles/10 mm	13.28	ACS Energy Lett. ²⁵
PES/Graphene/NiO _x /PVK/PCBM/AZO/Ag/AZO	14.00	1000 cycles/1.5%	12.60	Nano Energy ²⁶
MgF ₂ /PET/ITO/Nb ₂ O ₅ / Spiro-OMeTAD/Au	18.40	5000 cycles/4 mm	15.20	Adv. Mater. ²⁷
PET/ITO/SnO ₂ /PVK/ Spiro-OMeTAD/Au	15.22 (30 cm ²)	1800 cycles/-	10.66	Nat. Commun. ²⁸
PEN/ITO/WB-SnO ₂ /PVK/ Spiro-OMeTAD/Ag	18.00	1000 cycles/3 mm	10.98	Adv. Funct. Mater. ²⁹
PEN/ITO/SnO ₂ /PVK/ Spiro-OMeTAD/Ag	19.51	6000 cycles/8 mm	18.53	Adv. Energy Mater. ³⁰
PET/ITO/NiO _x /PVK/Bis-C ₆₀ /Ag	14.53	100 cycles/-	11.62	ACS Nano ³¹
PEN/ITO/HT-SnO ₂ /PVK/ PCBM/Ag	17.30	1000 cycles/14 mm	15.57	Adv. Funct. Mater. ³²
PET/ITO/NC-PEDOT:PSS/PVK PCBM/Ag	12.32 (1 cm ²)	1000 cycles/2 mm	11.46	Adv. Mater. ³³
PET/ITO/NiO _x /PVK/PCBM/ BCP/Ag	15.12	5000 cycles/2.5 mm	12.85	Adv. Funct. Mater. ¹⁹
PET/PEDOT:PSS:CFE/PVK/ PCBM/BCP/Ag	19.00	5000 cycles/3 mm	16.15	Joule ¹
PET/ITO/PEDOT:EVA/PVK/ PCBM/Ag	19.87 (1 cm²)	7000 cycles/3 mm	17.09	This work

- Hu, X., Meng, X., Zhang, L., Zhang, Y., Cai, Z., Huang, Z., Su, M., Wang, Y., Li, M., Yao, X., Wang, F., Ma, W., Chen, Y., Song, Y., A Mechanically Robust Conducting Polymer Network Electrode for Efficient Flexible Perovskite Solar Cells. *Joule* **3**, 2205-2218 (2019).
- Liu, C., Zhang, L., Zhou, X., Gao, J., Chen, W., Wang, X., Xu, B. Hydrothermally Treated SnO₂ as the Electron Transport Layer in High-Efficiency Flexible Perovskite Solar Cells with a Certificated Efficiency of 17.3%. *Adv. Funct. Mater.* **29**, 1807604 (2019).
- Yang, D., Yang, R., Wang, K., Wu, C., Zhu, X., Feng, J., Ren, X., Fang, G., Priya, S., Liu, S. High efficiency planar-type perovskite solar cells with negligible hysteresis using EDTA-complexed SnO₂. *Nat. Commun.* **9**, 3239 (2018).
- Wang, C., Zhao, D., Yu, Y., Shrestha, N., Grice, C. R., Liao, W., Cimaroli, A. J., Ellingson, R. J., Zhao, X., Yan, Y. Compositional and morphological engineering of mixed cation perovskite films for highly efficient planar and flexible solar cells with reduced hysteresis. *Nano Energy* **35**, 223-232 (2017).
- Wu, C., Wang, D., Zhang, Y., Gu, F., Liu, G., Zhu, N., Luo, W., Han, D., Guo, X., Qu, B., Wang, S., Bian, Z., Chen, Z., Xiao, L. FAPbI₃ Flexible Solar Cells with a Record Efficiency of 19.38% Fabricated in Air via Ligand and Additive Synergetic Process. *Adv. Funct. Mater.* **29**, 1902974 (2019).
- Li, H., Li, X., Wang, W., Huang, J., Li, J., Huang, S., Fan, B., Fang, J., Song, W. Ultraflexible and biodegradable perovskite solar cells utilizing ultrathin cellophane paper substrates and TiO₂/Ag/TiO₂ transparent electrodes. *Sol. Energy* **188**, 158-163 (2019).

25. Wang, C., Guan, L., Zhao, D., Yu, Y., Grice, C. R., Song, Z., Awni, R. A., Chen, J., Wang, J., Zhao, X., Yan, Y. Water Vapor Treatment of Low-Temperature Deposited SnO₂ Electron Selective Layers for Efficient Flexible Perovskite Solar Cells. *ACS Energy Lett.* **2**, 2118-2124 (2017).
26. Tran, V. D., Pammi, S. V. N., Park, B. J., Han, Y., Jeon, C., Yoon, S. G. Transfer-free graphene electrodes for super-flexible and semi-transparent perovskite solar cells fabricated under ambient air. *Nano Energy* **65**, 104018 (2019).
27. Feng, J., Zhu, X., Yang, Z., Zhang, X., Niu, J., Wang, Z., Zuo, S., Priya, S., Liu, s., Yang, D. Record Efficiency Stable Flexible Perovskite Solar Cell Using Effective Additive Assistant Strategy. *Adv. Mater.* **30**, 1801418 (2018).
28. Bu, T., Li, J., Zheng, F., Chen, W., Wen, X., Ku, Z., Peng, Y., Zhong, J., Cheng, Y. B., Huang, F. Universal passivation strategy to slot-die printed SnO₂ for hysteresis-free efficient flexible perovskite solar module. *Nat. Commun.* **9**, 4609 (2018).
29. Chen, C., Jiang, Y., Guo, J., Wu, X., Zhang, W., Wu, S., Gao, X., Hu, X., Wang, Q., Zhou, G., Chen, Y., Liu, J. M., Kempa, K., Gao, J. Solvent-Assisted Low-Temperature Crystallization of SnO₂ Electron-Transfer Layer for High-Efficiency Planar Perovskite Solar Cells. *Adv. Funct. Mater.* **29**, 1900557 (2019).
30. Huang, K., Peng, Y., Gao, Y., Shi, J., Li, H., Mo, X., Huang, H., Gao, Y., Ding, L., Yang, J. High-Performance Flexible Perovskite Solar Cells via Precise Control of Electron Transport Layer. *Adv. Energy Mater.* **9**, 1901419 (2019).
31. Zhang, H., Cheng, J., Lin, F., He, H., Mao, J., Wong, K. S., Jen, A. K.-Y., Choy, W. C. H. Pinhole-Free and Surface-Nanostructured NiO_x Film by Room-Temperature Solution Process for High-Performance Flexible Perovskite Solar Cells with Good Stability and Reproducibility. *ACS Nano* **10**, 1503-1511 (2016).
32. Liu, C., Zhang, L., Zhou, X., Gao, J., Chen, W., Wang, X., Xu, B. Hydrothermally Treated SnO₂ as the Electron Transport Layer in High-Efficiency Flexible Perovskite Solar Cells with a Certificated Efficiency of 17.3%. *Adv. Funct. Mater.* **29**, 1807604 (2019).
33. Hu, X., Huang, Z., Zhou, X., Li, P., Wang, Y., Huang, Z., Su, M., Ren, W., Li, F., Li, M., Chen, Y., Song, Y. Wearable Large-Scale Perovskite Solar-Power Source via Nanocellular Scaffold. *Adv. Mater.* **29**, 1703236 (2017).

Reviewer #5:

1. Comments: I believe that the author mentioned PEDOT:EVA adhesive property with PET. But my comment is that how ITO with PEDOT:EVA recovered the performance including solar cells and electrical property after bending. As the authors mentioned in revision letter, ITO is cracked after bending which mean that ITO composed In-Sn-O chemical structure is destroyed. However, the authors did not clearly mention this point. I believe that this point is very important issue for this paper and need to carefully address “why”.

Reply: Thanks for these comprehensive suggestions. We can understand reviewers’ concerns about these issues, which is important for a full explanation of the research contents. To further improve the research quality, we have made corresponding explanations and data supplements, and hoped that the reviewers can reconsider the possibility of this work being published on Nature Communications. Thank you very much for your reminding to improve the overall level for this manuscript.

First, we are grateful for the reviewers’ recognition of PEDOT:EVA cohesiveness. In order to prove the mechanical stability for ITO layer modified by PEDOT:EVA, we re-label the mechanical finite element simulation to reflect the actual situation more clearly (In the past simulation results, we only marked PEDOT:EVA and perovskite layer, we are very sorry for this). The corresponding results and specific simulation parameters are shown below and added to the manuscript and Supplementary Information.

Meanwhile, we also supplemented the Supplement Note 9 to illustrate this phenomenon. We construct a mechanical model with structure of ITO/HTL (PEDOT:EVA or PEDOT:PSS)/perovskite/PCBM/Ag. By combining the Young’s Modulus, poisson ratio and film thickness for different layers, the stress distribution of the whole devices in the bending process is deeply analyzed and simulated (As shown in Supplementary Fig. 39 and Supplementary Tab. 12 and Tab. 13). After applying the stress, we simulate the overall mechanical distribution for the corresponding ITO and perovskite layers respectively, and the results are shown in Supplementary Fig.40. It can be clearly found that PEDOT:EVA buffer layer can significantly reduce the overall stress distribution for the films in bending process, no matter for perovskite and ITO layers. This is due to the lower Young’s Modulus (The mechanical parameters and thickness of each layer were measured by the multifunctional mechanical tester and step profiler respectively) and density for the PEDOT:EVA film. The optimized mechanical parameters, compared with PEDOT:PSS film, provide a preliminary explanation for the improved mechanical stability of the ITO film.

Supplementary Fig. 39 Finite-elements simulation model of flexible PSCs upon PEDOT:EVA and PEDOT:PSS.

Supplementary Fig. 40 Finite-elements simulation for the perovskite and ITO layers upon PEDOT:EVA and PEDOT:PSS.

Supplementary Tab. 12 Mechanical parameters for finite element simulation of PEDOT:PSS-based perovskite devices.

Materials	Thickness (μm)	Young's modulus (Mpa)	Density (ρ , g cm^{-3})	Poisson ratio
ITO	0.185	840	6.80	0.25
PEDOT:PSS	0.035	471	1.39	0.32
PVK	0.35	843	4.1	0.27
PC ₆₁ BM	0.06	385	1.6	0.36
Ag	0.1	11435	10.5	0.38

Supplementary Tab. 13 Mechanical parameters for finite element simulation of PEDOT:EVA-based perovskite devices.

Materials	Thickness (μm)	Young's modulus (Mpa)	Density (ρ , g cm^{-3})	Poisson ratio
ITO	0.185	840	6.80	0.25
PEDOT:EVA	0.035	6	0.96	0.42
PVK	0.35	587	4.1	0.23
PC ₆₁ BM	0.06	385	1.6	0.36
Ag	0.1	11435	10.5	0.38

In addition, we explain the contribution of PEDOT:EVA layer to the mechanical stability of ITO layer from the mechanical structure analysis. As a conductive film, ITO conducts electricity by the carrier migration between the crystal lattice. When external forces are applied to the ITO film and obvious deformation occurs, once the inherent microstructure of the film is destroyed, the carrier migration channel will be affected, which is manifested as a sharp deterioration in electrical conductivity macroscopically. When the strain variable increases to a certain value, the structure of the ITO film will be seriously damaged, and its resistance value will have an obvious mutation, which is called **critical strain** (J. Mater. Sci-Mater. El. 2015, 26, 250-261; J. Disp. Technol. 2011,

7, 593-600; J. Disp. Technol. 2013, 9, 577-585). Previous reports have shown that the binding ability between the buffer layer and ITO layer will significantly affect the bending resistance for the brittle ITO layer (Org. Electron. 2010, 11, 670-676; Mater. Lett. 2013, 113, 182-185). As the adhesiveness recognized by reviewers, PEDOT:EVA shows impressive binding ability with perovskite and ITO layers through the adsorption and Van Der Waal force of PEDOT:EVA glue, which is also crucial for improving the mechanical stability of ITO layer. In order to explore the change of critical strain, we conducted in-situ measurements for the original ITO/PEDOT:EVA and ITO/PEDOT:PSS films (R_0) with the ITO/PEDOT:EVA and ITO/PEDOT:PSS films at different bending angles (R). The results are shown in Supplementary Fig. 41. For the ITO film coated with PEDOT:EVA, after 1500 bending cycles, the conductive property does not change significantly at the bending angle from 10 to 70 degree, which means the structure of ITO film is not damaged. However, for the ITO films coated with PEDOT:PSS, the electrical conductivity not only degrades obviously, but also increases with the increase of bending angle, suggesting the film structure is destroyed.

Supplementary Fig. 41 The conductivity of ITO film coated with PEDOT:EVA and PEDOT:PSS under different bending angles after 1500 cycles.

Moreover, in order to further explain the optimization of mechanical stability, we also calculate the **mechanical mismatch coefficient**. The damage condition of film mainly depends on the material property, but once the damage occurs, its further extension will depend on the release rate of strain energy in the film (Thin Solid Films 2001, 394, 201-205; Adv. Appl. Mech. 1992, 29, 63-191). Therefore, we analyze the above experiment results by the Dundurs coefficient (α and β).

$$\alpha = \frac{\bar{E} - \bar{E}_s}{\bar{E} + \bar{E}_s}$$

$$\beta = \frac{\bar{E} \left(\frac{1-2\nu_s}{1-\nu_s} \right) - \bar{E}_s \left(\frac{1-2\nu}{1-\nu} \right)}{2(\bar{E} + \bar{E}_s)}$$

Where E and E_s are the Young's Modulus of the PEDOT:EVA (PEDOT:PSS) and ITO layers, ν and ν_s are the poisson's ratio of the PEDOT:EVA (PEDOT:PSS) and ITO layers. In addition, \bar{E} and \bar{E}_s can be calculated by the following formula: $\bar{E} = E/(1-\nu^2)$ and $\bar{E}_s = E_s/(1-\nu_s^2)$. In the absence of energy loss, the value of β tends to zero and the α describes the mismatch degree of the Young's Modulus between the HTL and ITO layers, with values ranging from -1 to 1. The lower α value implies a worse mismatch degree and indicates a more effective inhibition for the microcracks. Due to the more optimized Young's Modulus of PEDOT:EVA, the ITO/PEDOT:EVA film (-0.703)

exhibits superior mechanical stability, compared with the ITO/PEDOT:PSS film (-0.506). We also calculated the strain energy stored in per unit width for the films before fracture (G_0) and the **strain energy release rate** (G_{ten}). The calculation of G_0 and G_{ten} are shown below.

$$G_0 = \frac{1}{2} \bar{E} h \varepsilon^2$$

$$G_{ten} = G_0 g(\alpha, \beta)$$

Where h is the film thickness, ε is the strain at a certain bending radius, $g(\alpha, \beta)$ is a function which is proportional to α (Thin Solid Films 2001, 394, 201-205; Adv. Appl. Mech. 1992, 29, 63-191; Thin Solid Films 2004, 460, 156-166). For the different films with similar thickness and bending radius, lower Young's Modulus means lower G_0 . In combination with the direct proportional relationship between $g(\alpha, \beta)$ and α , these will lead to a lower G_{ten} . Excessive strain energy release rate will not limit the extension of the damaged area in the film, on the contrary, the damaged area will be difficult to extend on the interface. This is consistent with the change of conductivity and surface morphology for the ITO/PEDOT:EVA and ITO/PEDOT:PSS films under 4500 bending cycles.

Therefore, we think this explanation is reasonable and hope that the reviewer can be satisfied. We have also added the corresponding description and supplement to the manuscript and Supplementary Information (Supplementary Note 9).

2. Comments: And from the DFT calculation, the author only compared to Pb atom with carboxyl group and sulfonic radical which indicates that EVA materials is only important for carboxylic group. This conclusion can be induced to any materials with carboxylic group can replace the PSS. Therefore, I believe that the authors need more clear evidence why EVA is important with PEDOT compared to other materials.

Reply: We are very grateful to the reviewers for their doubts about the results of DFT calculation and the scientific attitude.

Actually, we did not just compare to Pb atom with carboxyl group and sulfonic radical from the DFT simulation and the EVA materials is not only important for the carboxylic group. Meanwhile, we also suspect that any materials containing carboxyl groups are suitable for the synthesis of high-performance PEDOT materials. **In the DFT simulation process, we place four EVA molecules and two PSS molecules on the top of two surface terminations (PbI₂ or MAI terminations) to explore the interaction between perovskite and molecules.** To visually demonstrate this phenomenon, we conducted Car-parrinello Molecular Dynamics (CPMD) simulation and the results are shown in Fig. 2g, Supplementary Fig.6-9 and Supplementary Movie 3-6. In the 14 seconds of Supplementary Movie 3, it can be clearly found that the EVA molecules appear an obvious interaction with Pb atoms, which is derived from the interaction between the Pb atom and carboxyl group. However, the CPMD processes of EVA and MAI, PSS and PbI₂, PSS and MAI did not exhibit this tendency to form interactions (Supplementary Movie 4-6), demonstrating the regulatory mechanism of nucleation sites (In order to assist reviewers' reading, we cut these videos as shown in the following figure.). The interaction between the Pb atom and carboxyl group on EVA material is the explanation for the nucleation site regulation during the perovskite precursor crystallization.

Fig. The screenshots (14 seconds) of CPMD simulation videos for the EVA and PbI_2 , EVA and MAI, PSS and PbI_2 , PSS and MAI.

For other materials containing carboxyl group, appropriate recalculation and interpretation are necessary. And we did not intend to prove that any molecules with carboxyl group would have a similar effect, which is lax. **In previous reports, the CPMD simulation has been proved to be feasible by in-depth research for the interaction at the interface** (Science 2019, 365, pp. 473-478; Energy Environ. Sci. 2016, 9, 155-163; Phys. Chem. Chem. Phys. 2014, 16, 16137-16144; J. Phys. Chem. Lett. 2017, 8, 10, 2247-2252; Chem. Mater. 2015, 27, 13, 4885-4892; Adv. Funct. Mater. 2016, 26, 5297-5306).

The same phenomenon may not necessarily exist for other molecules with carboxyl groups, for **three main reasons**:

- (1) For other molecules containing carboxyl group, **exploring whether they interact with perovskite materials requires recalculation and simulation** based on the DFT molecular model, because different molecular structure, molecular conformation, functional group diversity, functional group number and steric hindrance effect will have important effects on simulation results, so it is not scientific to generalize the simulation results of EVA material to all molecules containing carboxyl groups.
- (2) Another feature of EVA is its **remarkable adhesiveness**, which is crucial for optimizing the mechanical stability of the PSCs and is also one of the starting points for selecting this material. We cannot determine whether other materials containing carboxyl group have this property, therefore, even if there is a certain interaction with perovskite layer, the perovskite devices based on other materials may not show excellent bending resistance.
- (3) At the same time, it is also worth noting whether other materials can **prepare the PEDOT ink by microemulsion method**. For other materials containing carboxyl group, actual synthesis or treatment is needed to prove whether they have the potential to be applied to hole transport layers. In addition, even if the PEDOT ink can be synthesized, the light transmittance, conductivity and adhesiveness for the films will need to be further evaluated.

To sum up, EVA can interact with Pb atom and shows the characteristics of both adhesiveness and synthesis of PEDOT ink, which can be sufficient to confirm that EVA is important with PEDOT compared to other materials. As for other materials containing carboxyl group mentioned by reviewers, we think this suspicion is not comprehensive enough. **This question is very important to highlight the advantages of EVA materials, and we have added the corresponding description to the revised manuscript (Page 3, paragraph 2, line 5) and Supplementary Note**

1. More importantly, these supplementary provide the guidance for the selection of similar materials in future reports, which is of great significance.

3. Comments: Finally, “vertebrae” means the similar issue with “recover” but the authors did not clearly mention “why”.

Reply: We are very grateful to the reviewer for the question about the bionic mechanism, and we have supplemented the manuscript and Supplementary Information (Supplementary Note 11) to make it easier to read.

The mechanism of “vertebrae” bionics mainly comes from two aspects (structure bionics and crystal bionics). **In terms of structure bionics**, articular cartilage is between the two vertebrae, either hyaline cartilage or fibrous cartilage, **which is similar to the characteristics of fibrous PEDOT:EVA layer** (Supplementary Fig. 37). Social activities of human are inseparable from the normal function of articular cartilage. Articular cartilage can ensure that the vertebrae are not damaged, one of the reasons is that the articular cartilage has the function of the force absorption. Articular cartilage can distribute the force evenly and enlarge the bearing surface. **This not only maximizes the mechanical load, but also protects the vertebrae from damage, which is due to the elasticity and adhesiveness of the articular cartilage. This feature is also similar to the PEDOT:EVA attribute** (Supplementary Fig. 23). Therefore, we mimic the structure of vertebrae/cartilage/vertebrae and design the flexural structure of ITO/PEDOT:EVA/PVK, which could be reasonable.

In terms of crystal bionics, we regard the nucleation process of perovskite precursor and the growth process of vertebrae as the bionic key point. Vertebrae growth is a mineralization process that involves four main steps (Nature 1988, 332, 6160; Adv. Mater. 2008, 20, 1333-1338; Adv. Mater. 2005, 17, 1461-1465; Adv. Mater. 2017, 29, 1605903):

- (1) Preorganization of organic matter: the **insoluble organic matter** in organisms forms an organized microreactive environment before mineral deposition, which determines the location of **inorganic matter nucleation** and the function of mineral formation.
- (2) Interface molecular recognition: under the control of the assembled organic macromolecules, inorganic materials nucleate at **the organic-inorganic interface** in solution by electrostatic force, chelation, hydrogen bond, van der Waals force, etc.
- (3) Growth modulation: the morphology, size, orientation and structure of the crystals are regulated by the organic matter of organisms during the growth of inorganic mineral phases, and **subunits** are initially assembled.
- (4) Crystal epitaxial growth.

It can be clearly found from the above description, for the growth of vertebrae, **the hydrophobic organic interface, organic-inorganic interaction sites and crystal growth** conditions are critical. This nucleation regulation process is very similar to the regulation mechanism of PEDOT:EVA HTL on nucleation in the manuscript. Therefore, we call the regulation of PEDOT:EVA on the perovskite crystallization as crystal bionics. Based on the above explanation, we believe that PEDOT:EVA material is feasible for the application of two bionic processes. The reviewers' suggestion helps us explain the mechanism of structure bionics and crystal bionics more clearly. **We have supplemented the corresponding description to the manuscript and a new Supplementary Note 11 has been also added in Supplementary Information to illustrate this bionic strategy.**

In fact, perovskite solar cells have a unique advantage over crystalline silicon solar cells in that they have the potential to be used in flexible photovoltaic devices. The power conversion efficiency for flexible perovskite solar cells prepared by printing technology has been preliminary satisfy the commercial standard currently, thus the further study of the new large-area printing technology, the design of large-area device structure, the establishment of large-area crystallization theory and the evaluation of the feasibility for the perovskite devices in low-power devices are one of the research focuses in next stage of perovskite solar cells. In this work, a large-area record perovskite solar cell was prepared by meniscus-printing method (the PCE of large-area PSMs devices has also been certified). On the other hand, we summarized a set of crystallization mechanism and research methods for the flexible perovskite solar cells, which is significant for the development of flexible devices. More importantly, these phenomena were confirmed by integrated of characterizations, mechanical and crystallization dynamics simulations. Therefore, we believe that this work can meet the requirements for publication in Nature Communications, and we also hope that reviewers can reevaluate the manuscript content. In the meantime, we would like to thank you for your questions on improving the quality for this manuscript.

REVIEWERS' COMMENTS:

Reviewer #4 (Remarks to the Author):

Authors addressed all the comments raised by the reviewers.

Response to reviewers' comments

Reviewer #4:

Comments: Authors addressed all the comments raised by the reviewers.

Reply: We thank the referee for a constructive review process.